# Host sirtuin 2 as an immunotherapeutic target against tuberculosis

**Ashima Bhaskar[1]\*, Santosh Kumar[2], Mehak Zahoor Khan[1], Amit Singh[3], Ved Prakash Dwivedi[2], Vinay Kumar Nandicoori[1]**

[1]Signal Transduction Laboratory 1, National Institute of Immunology, Aruna Asaf Ali Marg, New Delhi, India; [2]Immunobiology Group, International Centre for Genetic Engineering and Biotechnology, Aruna Asaf Ali Marg, New Delhi, India; [3]Department of Microbiology and Cell Biology, Centre for Infectious Disease Research, Indian Institute of Science, Bangalore, India

**Abstract** *Mycobacterium tuberculosis* (*Mtb*) employs plethora of mechanisms to hijack the host defence machinery for its successful survival, proliferation and persistence. Here, we show that *Mtb* upregulates one of the key epigenetic modulators, NAD+ dependent histone deacetylase Sirtuin 2 (SIRT2), which upon infection translocate to the nucleus and deacetylates histone H3K18, thus modulating the host transcriptome leading to enhanced macrophage activation. Furthermore, in *Mtb* specific T cells, SIRT2 deacetylates NFκB-p65 at K310 to modulate T helper cell differentiation. Pharmacological inhibition of SIRT2 restricts the intracellular growth of both drug-sensitive and resistant strains of *Mtb* and enhances the efficacy of front line anti-TB drug Isoniazid in the murine model of infection. SIRT2 inhibitor-treated mice display reduced bacillary load, decreased disease pathology and increased *Mtb*-specific protective immune responses. Overall, this study provides a link between *Mtb* infection, epigenetics and host immune response, which can be exploited to achieve therapeutic benefits.

**\*For correspondence:**
ashimabhaskar@gmail.com

**Competing interests:** The authors declare that no competing interests exist.

## Introduction

Tuberculosis (TB), a deadly disease caused by the intracellular pathogen *Mtb*, has existed since time immemorial and continues to remain one of the leading causes of mortality by a single infectious agent (*WHO, 2018*). Classic anti-TB therapy which comprises the administration of multiple anti-mycobacterial drugs, fails to provide complete sterilization in the host. Incessant rise in drug-resistant TB cases further highlights the failure of current anti-TB therapy which only focuses on targeting microbial pathways (*WHO, 2018*). The host immune system plays a pivotal role in the containment of the infection, while *Mtb* has evolved diverse strategies to avoid immune surveillance facilitating its survival, replication, and persistence in the host (*Korb et al., 2016*; *Mayer-Barber and Barber, 2015*). Our growing knowledge on host-pathogen interactions indicates that augmenting the current anti-TB therapy with host-directed strategies may result in enhanced bacterial clearance, shorter treatment times, reduced tissue damage, a decline in drug-resistant strains and a lower risk of relapse (*Palucci and Delogu, 2018*).

For its enormous success as an intracellular pathogen, *Mtb* skews multiple host pathways in its favor. For example, *Mtb* is known to restrict the killing capacity of macrophages by inhibiting host generated oxidative stress, apoptosis and multiple stages of autophagy (*Krakauer, 2019*; *Lam et al., 2017*). It also influences the adaptive immune response by promoting the secretion of T helper 2 (Th2) polarizing cytokines (*Bhattacharya et al., 2014*). Moreover, *Mtb* infection significantly changes the transcriptional landscape of host cells (*Roy et al., 2018*) by secreting a plethora of virulence factors to carry out these functions. It also hijacks the function of several host genes for its

gain (*Hawn et al., 2013*). Yet another mechanism has been uncovered recently (*Hamon and Cossart, 2008*), wherein intracellular pathogens remodel the host chromatin for their persistence.

A balance between histone acetylation and deacetylation carried out by histone acetyltransferases (HATs) and histone deacetylases (HDACs), respectively, play a crucial role in the regulation of gene expression. Till date, few bacteria have been reported to modulate the levels of acetylated histones. *Listeria monocytogenes*, a food-borne pathogen, dephosphorylates H3 and deacetylates H4 in epithelial cells through the action of a secreted virulence factor: listeriolysin O (*Hamon et al., 2007*). *Anaplasma phagocytophilium*, which causes human granulocytic anaplasmosis has been shown to activate the expression of HDAC1 and HDAC2 leading to transcriptional repression of key immunity genes (*Garcia-Garcia et al., 2009*). The influence of *Mtb* infection on histone modifications and chromatin remodeling is still in its infancy. It has been shown that *Mtb* inhibits the expression of IFNγ-induced genes including CIITA, CD64, and HLA-DR through histone deacetylation (*Kincaid and Ernst, 2003*; *Wang et al., 2005*). Moreover, broad-spectrum HDAC inhibitors enhance the antimycobacterial potential of host cells (*Moreira et al., 2020*).

The class III HDACs, or sirtuins (SIRT1-7) are homologous to the yeast Sir2 family of proteins and require NAD$^+$ as a cofactor that links their enzymatic activity to the energy state of a cell. Thus far, very few studies have demonstrated the role of sirtuins in bacterial pathogenesis. Recent works emphasize the importance of SIRT1 and SIRT2 in the progression of bacterial infections (*Cheng et al., 2017*; *Eskandarian et al., 2013*; *Gogoi et al., 2018*). Despite enhanced phagocytosis in SIRT2-deficient macrophages (*Ciarlo et al., 2017*), myeloid-specific SIRT2 deficiency fails to control *Mtb* growth in mice (*Cardoso et al., 2015*).

SIRT2 primarily a cytoplasmic protein, is known to shuttle into the nucleus during mitosis (*North and Verdin, 2007*) where it regulates chromosome condensation. Mounting evidence suggests the role of SIRT2 in cell cycle regulation, tumorigenesis, neurodegeneration, cellular metabolism and energy homeostasis (*Gomes et al., 2015*).

In the present study, we attempt to decipher the role of SIRT2 in TB pathogenesis using chemical inhibition of SIRT2. We show that *Mtb* infection leads to upregulation and nuclear translocation of SIRT2 which induce consequential changes in histone acetylation, cellular signaling and transcriptional profile of the infected host macrophages and *Mtb*-stimulated T cells. Inhibition of SIRT2 activity restricted *Mtb* growth and reduced disease pathology in infected mice. SIRT2 inhibition triggered the induction of host protective immune responses, enhanced the efficacy of front-line anti-TB drug isoniazid (INH) and showed protection against drug-resistant strains. Thus, the study unravels a novel epigenetic mechanism employed by *Mtb* to recondition the host defense system, which can be exploited toward designing host-directed adjunct therapy.

## Results

### SIRT2 inhibition restricts *Mtb* growth ex vivo

To delve into the epigenetic modifications induced by *Mtb* to evade host defence system, we evaluated the expression of all known HDACs in THP1 cells 24 hr post *Mtb* infection (pi) through microarray analysis (*Mehta et al., 2016*), Gene Expression Omnibus series accession number **GSE65714**. Among a total of 18 HDACs, SIRT2 and HDAC6 were significantly upregulated whilst SIRT1 and HDAC9 were downregulated (*Figure 1A*; *Mehta et al., 2016*). This is in accordance with a previous report indicating the *Mtb* mediated downregulation of SIRT1 (*Cheng et al., 2017*). The increase in the expression of SIRT2 following *Mtb* infection was further confirmed in mouse peritoneal macrophages by qRT-PCR (*Figure 1B*) and intracellular staining with α-SIRT2 followed by flow cytometry (*Figure 1C and D*). To uncover the biological relevance of *Mtb* mediated SIRT2 upregulation, we analyzed the intracellular survival of *Mtb* in peritoneal macrophages upon addition of AGK2, a well-established SIRT2 inhibitor (*Outeiro et al., 2007*). Flow cytometric analysis of peritoneal macrophages infected with GFP expressing bacteria revealed significantly reduced *Mtb* growth (*Figure 1E*). Similarly, CFU enumeration indicated a significant reduction in the bacillary load upon AGK2 treatment (*Figure 1F*). Unlike Ciarlo et al, we did not observe any significant difference in the uptake of *Mtb* by macrophages treated with AGK2 (*Figure 1E and F*). To further ascertain the effects of SIRT2 downregulation on *Mtb* survival, we transfected RAW 264.7 macrophages with mouse shRNA-directed against SIRT2. Among the four shRNAs tested, shRNA three and shRNA five

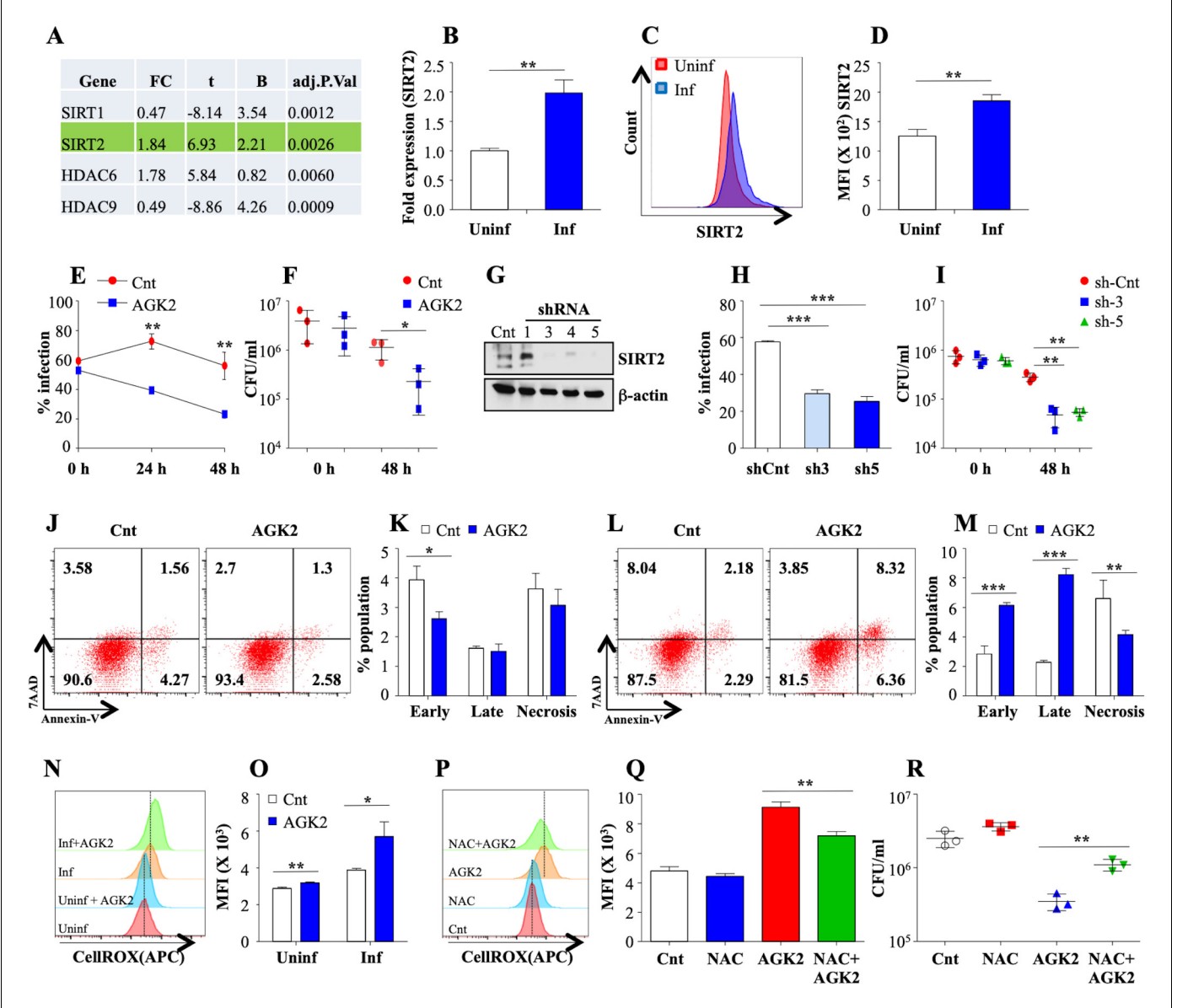

**Figure 1.** SIRT2 aids in mycobacterial survival ex vivo. (A) Analysis of microarray data from *Mehta et al., 2016* reveals differential expression of SIRT1, SIRT2, HDAC6 and HDAC9 in THP1 cells 24 hr pi. FC: Fold Change in the gene expression in infected THP1 cells as compared to uninfected control, t: moderated t-statistics, B: B-statistics or log-odds that the gene is differentially expressed, adj.p.val: p-value after adjustment for multiple testing. (B) Mouse peritoneal macrophages were infected with *Mtb* laboratory strain *H37Rv* for 24 hr and SIRT2 expression was assessed through RT-PCR. (C–D) Mouse peritoneal macrophages infected with *Mtb* for 24 hr were stained with anti SIRT2 antibody followed by FACS analysis. (E and F) Mouse peritoneal macrophages, pre-treated with AGK2 for 2 hr, were infected with GFP expressing H37Rv and maintained in 10 μM of AGK2 for 48 hr. (E) Percentage of infected cells as analyzed by flow cytometry. (F) In a parallel experiment, cell lysates were plated for bacterial CFU at 0 hr and 48 hr pi. (G–I) RAW 264.7 macrophages were transfected with shRNAs specific for SIRT2. (G) 48 hr post transfection, SIRT2 protein levels were assessed by western blot. Transfected cells were infected with GFP expressing H37Rv for 48 hr. (H) Percentage of infected cells as analyzed by flow cytometry at 48 hr pi. (I) In a parallel experiment, cell lysate was plated for enumeration of CFU. (J–M) Representative dot plots and bar graphs to show the percentage of necrotic and apoptotic cell populations in the uninfected (J and K) and *Mtb* infected (L and M) peritoneal macrophages with or without AGK2 treatment as analyzed by FACS analysis. (N and O) Representative overlay plots and quantification of cellular ROS measured by staining cells with CellROX followed by flow cytometry. Mouse peritoneal macrophages treated with AGK2 (10 μM), NAC (10 mM) or both were infected H37Rv for 48 hr. (P and Q) Representative overlay plots and bar graph depicting cellular ROS in these cells 48 hr pi. (R) In a parallel experiment, cell lysates were plated for CFU enumeration. Data shown is representative of at least two independent experiments performed in triplicates. The data values represent mean ± SD (n = 3). *p<0.05, **p<0.005, ***p<0.0005.

The online version of this article includes the following source data for figure 1:

**Source data 1.** SIRT2 inhibition enhances anti-mycobacterial potential of host macrophages.

showed maximum downregulation of SIRT2 (*Figure 1G*). Similar to AGK2 treatment, bacterial survival was markedly attenuated in RAW 264.7 macrophages transfected with shRNA 3 and 5 (*Figure 1H and I*). SIRT2 is known to modulate the expression of antioxidant genes thereby influencing cellular redox homeostasis (*Gomes et al., 2015*). Furthermore, inhibition or downregulation of SIRT2 leads to the induction of apoptosis in many tumor cell lines (*Kozako et al., 2018*; *Zhang et al., 2016*). Since both apoptosis and ROS constitute one of the major innate defense strategies employed by macrophages against *Mtb* (*Behar et al., 2011*) and SIRT2 has been previously shown to modulate the induction of apoptosis and intracellular ROS in different disease settings (*Xu et al., 2019*), we analyzed these parameters upon SIRT2 inhibition during *Mtb* infection. While AGK2 treatment has minimal effect on uninfected macrophages (*Figure 1J and K*), its treatment significantly increased the percentage of apoptotic cells in *Mtb*-infected macrophages with a concomitant decrease in the necrotic population (*Figure 1L and M*). Interestingly AGK2 treatment also led to a significant increase in the intracellular ROS levels following *Mtb* infection with minimal effect on uninfected cells (*Figure 1N and O*). Moreover, treatment of cells with an anti-oxidant, N-acetyl cysteine (*Bhaskar et al., 2015*) significantly decreased the anti-mycobacterial effect of AGK2 (*Figure 1P–R*) indicating that ROS generation is one of the mechanisms by which SIRT2 inhibition restricts bacillary growth. Collectively, these results indicate a role for SIRT2 in assisting mycobacterial survival inside the host.

## *Mtb* Infection triggers SIRT2 migration into the nucleus

SIRT2 predominantly resides in the cytoplasm where it targets cytosolic proteins to exert its functions. However, it is known to migrate to the nucleus during the G2/M transition and certain bacterial infections (*Eskandarian et al., 2013*; *Vaquero et al., 2006*). To address whether similar translocation is promoted during *Mtb* infection, we determined the site of action of SIRT2 through immunofluorescence. Interestingly, while SIRT2 presence was observed in both the cytosol and the nucleus of uninfected macrophages, SIRT2 predominantly localized to the nucleus following *Mtb* infection as early as 4 hr pi (*Figure 2A*). SIRT2 migration into the nucleus was also confirmed by performing immunoblotting at 24 hr pi (*Figure 2B*). Significant increase in the overall expression of SIRT2 at protein level post *Mtb* infection correlated, with our qRT-PCR analysis (*Figures 1B* and *2B*). Since SIRT2 modulates the acetylation levels of histone H3K18 upon infection with *Listeria monocytogenes*, we analyzed H3K18 acetylation upon *Mtb* infection and AGK2 treatment in mouse peritoneal macrophages. AGK2 treatment had minimal effect on the acetylation levels of H3K18 in uninfected cells. However, there was a significant increase in H3K18ac levels in infected macrophages treated with AGK2, indicating that *Mtb*-induced nuclear localization of SIRT2 results in deacetylation of histone H3K18 (*Figure 2C and D*). α-tubulin is the major cytosolic target of SIRT2 which deacetylates it at lysine-40 to attenuate microtubule stability (*North et al., 2003*). In order to evaluate the cytosolic activity of SIRT2, we checked the acetylation levels of α-tubulin following *Mtb* infection and upon SIRT2 inhibition. AGK2 treatment led to a comparable increase in the acetylated α-tubulin levels in uninfected and infected macrophages indicating the presence of cytosolic SIRT2 activity under both the conditions (*Figure 2E*). However, increased nuclear localization of SIRT2 after *Mtb* infection may contribute to the enhanced acetylated α-tubulin levels in the *Mtb* infected cells as compared to their uninfected counterparts (*Figure 2E*). SIRT2 modulates intracellular signaling by manipulating the activation of key Ser/Thr kinases in the host cells (*Jing et al., 2007*; *Kosciuczuk et al., 2019*; *Li et al., 2018*). Thus we evaluated if SIRT2 inhibition had any impact on intracellular signaling. We measured the phosphorylation of ERK1/2 and p38 in peritoneal macrophages 2 hr after *Mtb* infection by immunoblotting. Interestingly, *Mtb* induced phosphorylation of ERK1/2 and p38 was decreased by AGK2 treatment (*Figure 2F*). Since SIRT2 inhibition strongly influenced the intracellular signaling and histone deacetylation, both of which ultimately lead to changes in cellular gene expression, we hypothesize that SIRT2 influences the intracellular *Mtb* growth by modulating the host gene expression.

## SIRT2 induce global host gene expression changes during *Mtb* infection

To further comprehend the underlying mechanisms involved in restricting *Mtb* growth upon SIRT2 inhibition, we carried out transcriptome analysis by performing RNAseq on uninfected and *Mtb*-infected peritoneal macrophages with and without AGK2 treatment (U: Uninfected macrophages,

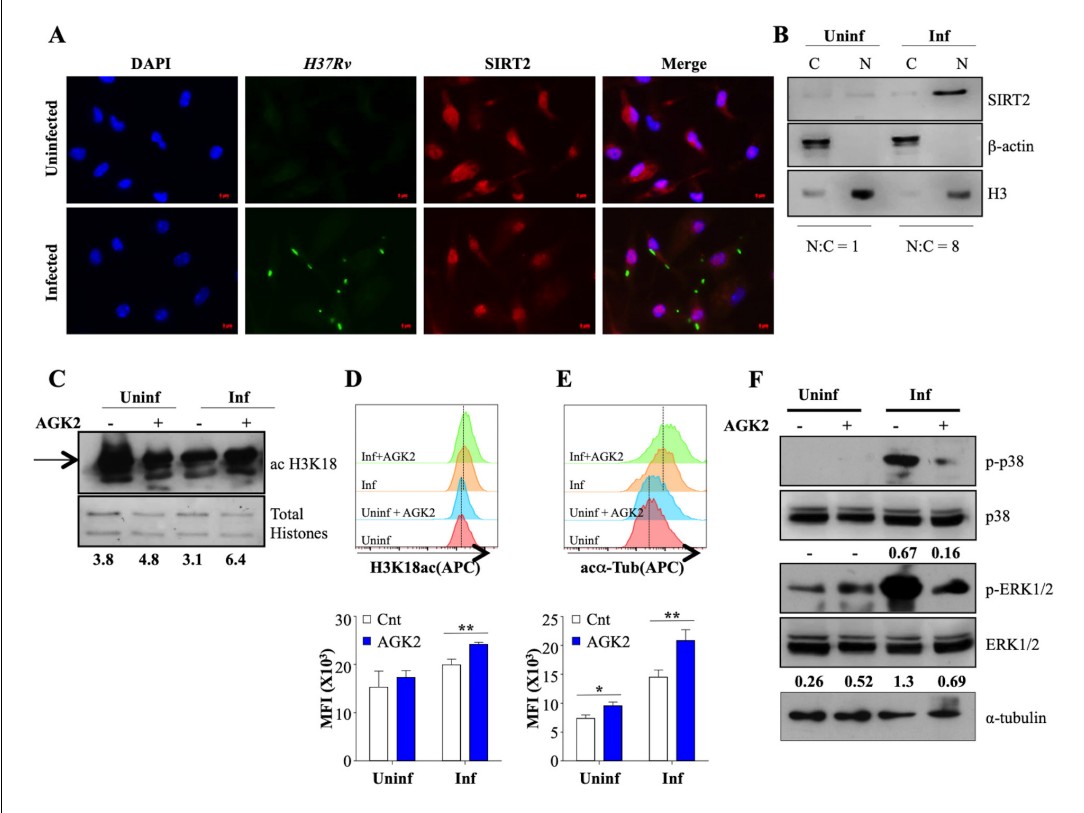

**Figure 2.** SIRT2 translocate to the nucleus and modulates histone deacetylations and cellular signaling during *Mtb* infection. (A) Endogenous SIRT2 was detected by immunofluorescence in mouse peritoneal macrophages uninfected or infected with *H37Rv* for 4 hr. (B) Uninfected macrophages or macrophages infected with H37Rv for 24 hr were fractionated for cytosol and nucleus followed by immunoblotting for the indicated proteins. Mouse peritoneal macrophages, pre-treated with AGK2 for 2 hr were infected *H37Rv* followed by AGK2 treatment for 24 hr. Acetylation levels of H3K18 were checked by (C) immunoblotting and (D) intracellular staining followed by flow cytometry. (E) Acetylated α-tubulin levels in uninfected and *Mtb*-infected cells with or without AGK2 treatment at 24 hr pi. (F) Mouse peritoneal macrophages, pre-treated with AGK2 for 2 hr were infected with *H37Rv* for 2 hr. Phosphorylation status of the indicated proteins was checked by immunoblotting. Data shown is representative of at least two independent experiments performed in triplicates. Each bar represents mean ± SD (n = 3). *p<0.05, **p<0.005.

The online version of this article includes the following source data for figure 2:

**Source data 1.** SIRT2 migrates to the nucleus upon *Mtb* infection.

UA: Uninfected macrophages treated with AGK2; I: Macrophages infected with *Mtb*; IA: AGK2 treated infected macrophages). AGK2 treatment in uninfected cells did not alter the gene expression in a significant manner which is consistent with our earlier results wherein no effect on apoptosis, ROS levels, histone deacetylations and cellular signaling was observed in uninfected cells treated with AGK2 (*Figures 1K, O*, *2D and F*). In contrast, AGK2 treatment significantly affected the gene expression profile induced upon *Mtb* infection. In total, *Mtb* infection led to the differential expression of 992 genes which is in agreement with the previous findings (*Mehta et al., 2016*; *Roy et al., 2018*). Genes that remained unaltered after AGK2 treatment were categorized as SIRT2 independent (414 genes) and the rest as SIRT2 dependent (578 genes) (*Figure 3A*). SIRT2-dependent genes can be further classified into six categories. Group 1: Decreased expression in I with no change in IA. Group 2: No change in expression in I and increased expression in IA. Group 3: Enhanced gene expression in IA as compared to I. Group 4: Decreased expression in IA with no change in I. Group 5: Increased expression in I with no change in IA. Group 6: Enhanced gene expression in I as compared to IA (*Figure 3A*). KEGG pathway analysis on SIRT2-dependent genes belonging to these groups (expect group 5) revealed that a significant number of pathways that are known to play a crucial role in combating TB were affected by AGK2 treatment (*Figure 3B*). For instance, genes involved in antigen presentation, Th17 cell differentiation, cytokine-cytokine receptor interaction, NF-κB signaling, tuberculosis, etc were significantly upregulated while PI3K/Akt signaling pathway

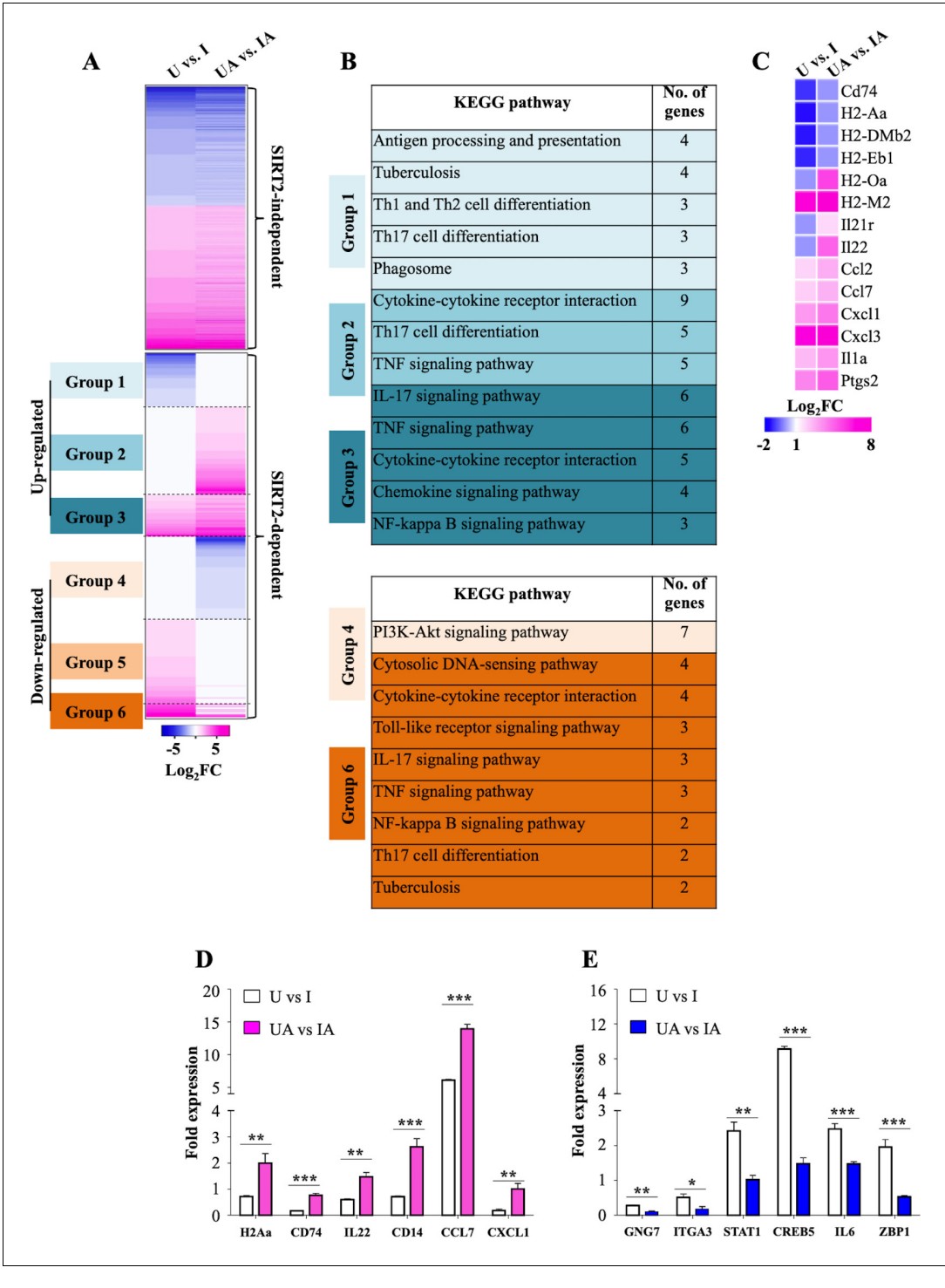

**Figure 3.** SIRT2 modulates host gene transcription during *Mtb* infection. (A) Heatmap interpretation of gene expression changes (Log2 fold) as determined by RNAseq analysis of peritoneal macrophages infected with *Mtb* for 24 hr. Pink depicts upregulation while blue represents repression. (B) KEGG analysis of molecular signaling pathways affected by AGK2 treatment. (C) Heatmap representing the expression profile of the genes involved in T cell activation. qRT-PCR of candidate genes which are either (D) up- or (E) downregulated in SIRT2 dependent manner. . U: Uninfected. UA: Uninfected treated with AGK2. I: Infected. IA: Infected with AGK2 treatment. RNAseq was performed in three biological replicates. Each bar in (D) and (E) represents mean ± SD (n = 3). *p<0.05, **p<0.005, ***p<0.0005.

The online version of this article includes the following source data for figure 3:

**Source data 1.** SIRT2 modulates transcriptional landscape of Mtb-infected macrophages.

was downregulated in SIRT2 inhibited samples (*Figure 3B*). Interestingly, the expression of genes involved in T cell activation was enhanced in SIRT2-dependent manner (*Figure 3C*). RNAseq data were validated by qRT-PCR of two genes from each group (*Figure 3D and E*). These results reiterate the existence of a novel anti-defense strategy employed by *Mtb* wherein it hijacks host SIRT2 to rewire the host transcriptional landscape for its survival.

## SIRT2 inhibition activates T-cell-mediated immunity via enhanced antigen presentation

Our RNAseq data revealed SIRT2-dependent expression of many genes which drives the production of cytokines and activation of T cells (*Figure 3C*). Innate and adaptive immunity play a crucial role in shaping the disease outcome in case of TB. Previous studies have revealed the influence of SIRT2 on the production of pro-inflammatory cytokines (*Lo Sasso et al., 2014*; *Yuan et al., 2016*). To replicate the in vivo conditions aimed at examining the effect on SIRT2 inhibition on the activation of macrophages and T cells, we performed co-culture experiments wherein splenocytes isolated from *Mtb* infected mice and peritoneal macrophages from naïve animals were either pre-treated separately with AGK2 prior to co-culture or infected peritoneal macrophages were co-cultured with splenocytes in the presence of AGK2 (*Figure 4A*). 48 hr post co-culture, the properties of macrophages and T cells were assessed by staining with antibodies against different surface markers and cytokines followed by FACS analysis. SIRT2 inhibition led to a marked increase in the expression of monocyte differentiation markers, CD11b and CD11c (*Figure 4B–D*). AGK2 treatment resulted in the increased expression of MHCII and CD80 co-stimulatory molecules which are known to stimulate T cells (*Figure 4B, E and F*). Interestingly, the expression of CD11b, CD11c and co-stimulatory markers CD80 and MHCII was significantly increased even in the condition where only the T cells were pre-treated with AGK2 (*Figure 4B–F*). With no change in the percentage of CD4$^+$ T cells (*Figure 4G and H*), we observed an increase in their activation as evidenced by an increased expression of T cell activation markers CD69 and CD25 (*Figure 4I and J*). A similar trend was observed in CD8$^+$ T cells (*Figure 4K–M*). Furthermore, SIRT2 inhibition increased the expression of pro-inflammatory cytokines IFNγ and IL17 by CD4$^+$ and CD8$^+$ T cells (*Figure 4N–Q*). However, SIRT2 inhibition in macrophage alone failed to drive strong Th1 and Th17 responses in CD4$^+$ and CD8$^+$ T cells while co-treatment of macrophages and T cells with AGK2 had a synergistic effect as IFNγ and IL17 production was significantly higher when splenocytes were co-cultured with macrophages in the presence of AGK2 (*Figure 4N–Q*). Similarly, the treatment of AGK2 in splenocytes alone or in both the cell types led to a significantly higher reduction in bacterial survival as compared to the SIRT2 inhibition in macrophages alone (*Figure 4R*). Overall, these results implicate a role for SIRT2 in modulating host protective T cell responses.

## SIRT2 deacetylates NFκB p65 in *Mtb*-specific T cells

The biological consequences of SIRT2 inhibition in T cells remain largely unexplored. In order to assess the SIRT2 activity in T cells, we isolated splenocytes from naive and *Mtb*-infected mice and stimulated them with *Mtb* complete soluble antigen (CSA) for 24 hr in the presence or absence of AGK2 (*Figure 5A*). Interestingly, we observed a significantly increased expression of SIRT2 in CD3$^+$ lymphocytes isolated from the spleen of *Mtb*-infected animals (*Figure 5B and C*). Replicating the profile obtained in *Mtb*-infected macrophages, SIRT2 inhibition by AGK2 modulated the acetylation levels of H3K18 in these cells (*Figure 5D*). Furthermore, SIRT2 deacetylated NFκB p65 subunit at K310 in *Mtb*-specific CD3$^+$ lymphocytes (*Figure 5E*).

## SIRT2 inhibition reduces the bacterial burden in mice

The ex vivo data encouraged us to replicate our findings in the host, towards which we infected a group of C57BL/6 mice with a low dose of *H37Rv* through aerosol. At 15 days pi, the mice were treated with AGK2 followed by an enumeration of bacterial burden and analysis of disease pathology at 45 days pi (*Figure 6A*). Histopathology analysis of infected lungs showed an overall decrease in lung inflammation after SIRT2 inhibition (*Figure 6B–D*). These data were further strengthened by the CFU analysis wherein the bacterial burden was significantly diminished in the lungs, spleen and liver of mice treated with AGK2 as compared with the control mice (*Figure 6E–G*). These results reiterated the inhibitory effects of SIRT2 deficiency on mycobacterial growth.

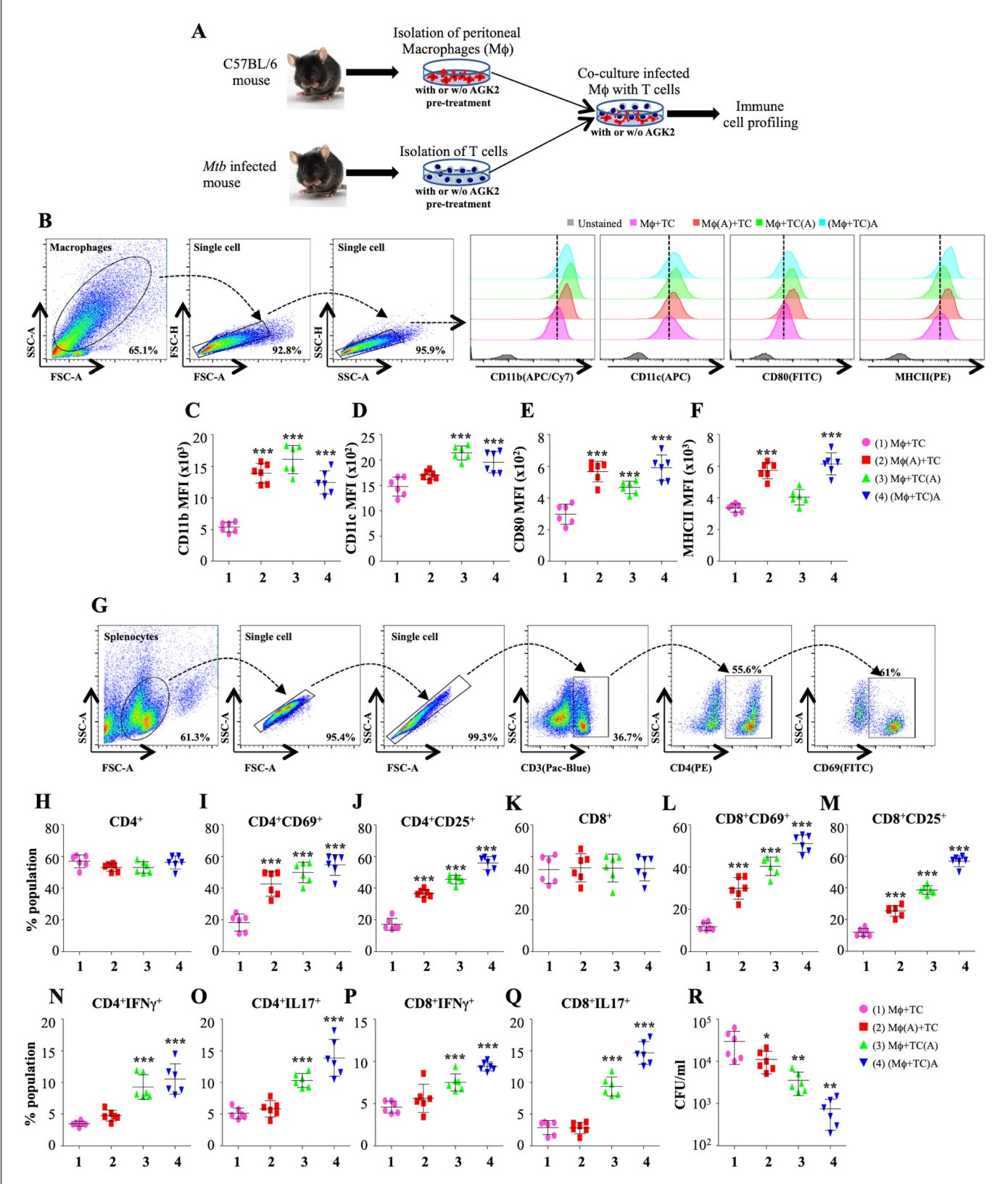

**Figure 4.** SIRT2 inhibition activates macrophages and T cells to induce pro-inflammatory response. (**A**) Schematic representation of the experimental plan (for details see Materials and methods). 48 hr post co-culture, macrophages were stained with CD11b (APC/Cy7), CD11c (APC), CD80 (FITC) and MHC-II (PE) followed by flow cytometry analysis. (**B**) Gating strategy employed and representative overlay plots to depict macrophage activation. *Figure 4 continued on next page*

**Figure 4 continued**

Expression of (**C**) CD11b, (**D**) CD11c, (**E**) CD80 and (**F**) MHCII on the surface of peritoneal macrophages. (**G**) Gating strategy for T cell activation. Splenocytes were stained for surface markers CD3 (Pacific Blue), CD4 (PE), CD8 (APC/Cy7), CD69 (FITC) and CD25 (APC) followed by flow cytometry. Percentage of (**H**) CD4$^+$, (**I**) CD4$^+$CD69$^+$, (**J**) CD4$^+$CD25$^+$, (**K**) CD8$^+$, (**L**) CD8$^+$CD69$^+$ and (**M**) CD8$^+$CD25$^+$ T cells. For cytokine analysis, splenocytes were stained with CD3 (Pacific Blue), CD4 (PE), CD8 (APC/Cy7), IFNγ (APC) and Il17 (PE/cy7). (**N and O**) Intracellular cytokines (IFNγ and IL17) expressed by CD4$^+$ T cells. (**P and Q**) CD8$^+$ T cells expressing IFNγ and IL17. (**R**) CFU at 48 hr post co-culture. Combined data from two independent experiments performed in triplicates is shown. Each bar represents mean ± SD (n = 6). Statistical analysis performed using one-way ANOVA followed by Tukey post hoc test. *p<0.05, **p<0.005, ***p<0.0005.

The online version of this article includes the following source data for figure 4:

**Source data 1.** SIRT2 inhibition leads to the activation of macrophages and T cells upon *Mtb* infection.

In contrast to the previous report where myeloid-specific deletion of Sirt2 failed to display long-lasting changes in TB control (*Cardoso et al., 2015*), administration of AGK2 significantly diminished the bacterial growth at early time points as well as at chronic late phase of infection. We speculate that the discrepancy in observed results is due to a fundamental difference between these two studies. While Cardoso et al used myeloid-specific Sirt2 KO mice, we employed the use of AGK2 which results in a more generic inhibition of SIRT2 activity thus emphasizing the importance of cells other than macrophages in containing *Mtb* infection. To corroborate our ex vivo results which emphasized the importance of SIRT2 inhibition in T cells for restricting intracellular *Mtb* growth, we performed infection experiments in *Rag1$^{-/-}$* mice that lack mature B and T cells. AGK2 treatment in these immunodeficient mice did not alter the bacterial survival as compared to the control group (*Figure 6H–J*) indicating that SIRT2 inhibition in T cells is indispensable in restricting *Mtb* growth inside the host.

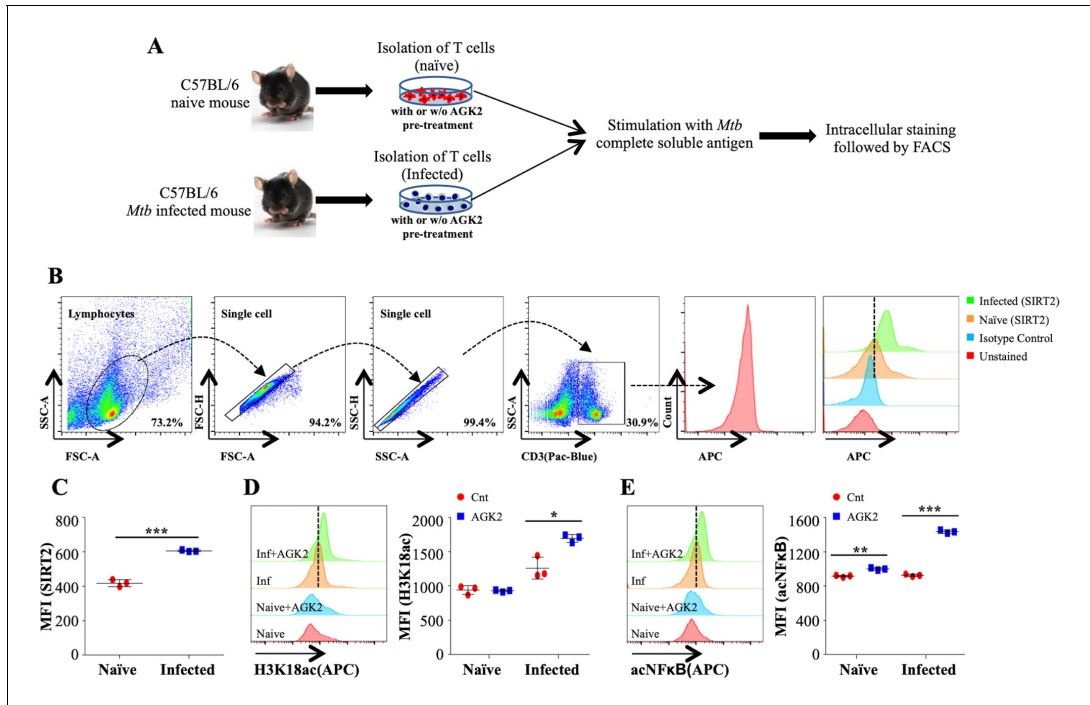

**Figure 5.** SIRT2 targets histone and non-histone proteins in *Mtb*-specific T-cells to ameliorate anti-mycobacterial host immune responses. (**A**) Schematic representation of the experimental plan. (**B**) Gating strategy utilized to analyze the intracellular levels of (**C**) SIRT2, (**D**) H3K18ac and (**E**) acetylated NFκB p65. Data shown is representative of at least two independent experiments each performed in triplicates. Data is represented as mean ± SD (n = 3). *p<0.05, **p<0.005, ***p<0.0005.

The online version of this article includes the following source data for figure 5:

**Source data 1.** SIRT2 deacetylates H3K18 and NFκB-p65 in *Mtb*-specific T cells.

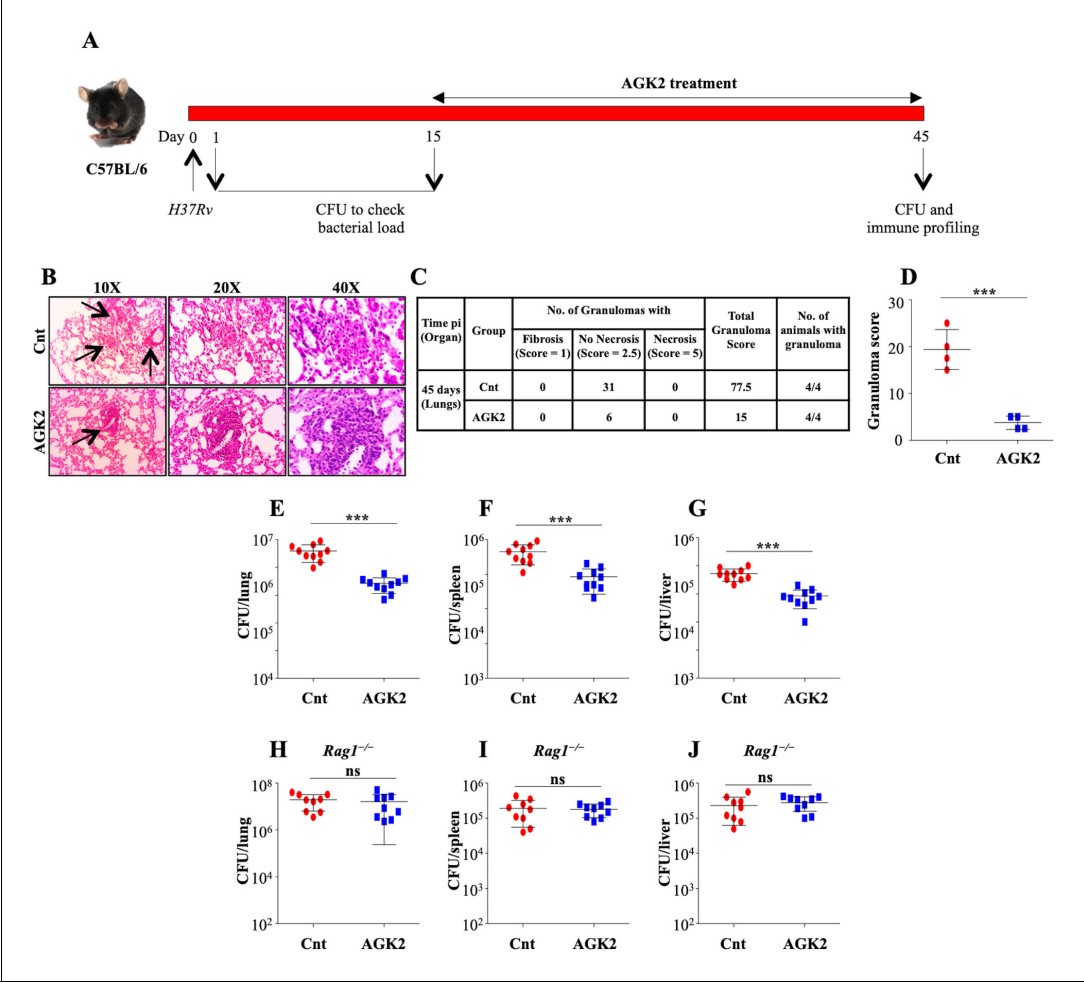

**Figure 6.** SIRT2 inhibition restricts mycobacterial growth in vivo. (**A**) Schematic representation of the murine model of infection. A group of C57BL/6 mice were infected with low dose of *H37Rv*. After 15 days of disease establishment, mice were either left untreated or were treated with AGK2 (20 mg/kg) for 30 days. (**B**) Histopathological analysis of infected lungs with arrows indicating the granulomatous lesions. (**C and D**) Quantification of the number of granulomas (granuloma score) observed in the infected mice. (**E–G**) Bacterial load in the lungs, spleen and liver of mice at 45 days pi. (**H–J**) A group of *Rag1⁻/⁻* mice were infected with low dose of H37Rv followed by treatment with AGK2 (20 mg/kg) for 30 days. The graphs represent bacterial burden in the lungs, spleen and liver of *Rag1⁻/⁻* mice. E-J represents combined data from two independent experiments with four to five mice in each group. Granuloma score was obtained from the lungs of four mice per group. Data is represented as mean ± SD. *p<0.05, **p<0.005, ***p<0.0005. The online version of this article includes the following source data for figure 6:

**Source data 1.** AGK2 treatment reduces bacterial burden in murine model of TB.

## AGK2 treatment activates host protective immune response against *Mtb*

The unrivalled ability of *Mtb* to survive within the host ensues from its strategies to evade the host immune defense system. Antigen-presenting cells (APCs) and Th1 cells are the key players contributing to the immunological control of *Mtb* infection. Encouraged by the results presented above, we profiled different immune cells in the lungs and the spleen of infected mice treated with AGK2 at 30 days post-treatment to fathom out the ability of AGK2 in modulating host protective adaptive immune responses for enhanced bacterial clearance. Increased percentage of CD11b⁺, CD11c⁺, CD80⁺ and MHCII⁺ cells, which are involved in the stimulation of adaptive immune responses, was observed in the lungs (*Figure 7A–E*) and the spleen of AGK2-treated mice (7F-7I). Similar to our ex vivo results, we observed an increased activation of CD4⁺ and CD8⁺ T cells in the lungs (*Figure 8A–G*) and the spleen (*Figure 8H–M*) of infected animals after AGK2 treatment. Moreover, the Th1 and Th17 bias in the immune response was evidenced by a prominent increase in the population of INFγ

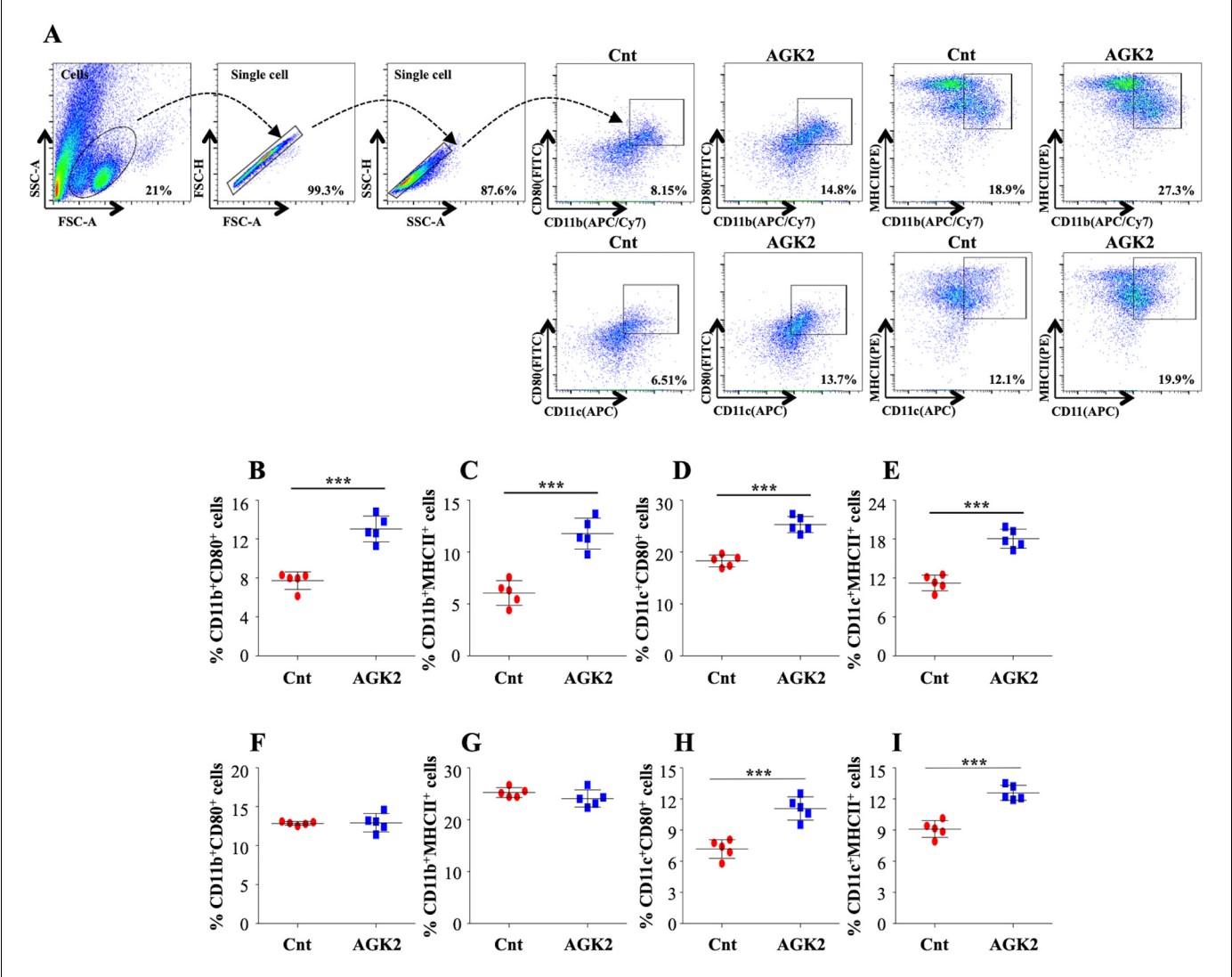

**Figure 7.** Inhibition of SIRT2 activity enhances macrophage stimulation in the lungs and spleen of infected animals. (A) After overnight stimulation with CSA, the cells isolated from the lungs of infected C57BL/6 animals were surface stained with CD11b (APC/Cy7), CD11c (APC), CD80 (FITC) and MHC-II (PE) followed by flow cytometry analysis. Gating strategy employed and flow cytometry dot plots of CD11b⁺CD80⁺, CD11b⁺MHCII⁺, CD11c⁺CD80⁺, CD11c⁺MHCII⁺ cells in the lungs of infected mice (B–E) Percentage of (B) CD11b⁺CD80⁺, (C) CD11b⁺MHCII⁺, (D) CD11c⁺CD80⁺ and (E) CD11c⁺MHCII⁺ cells in the lungs. (F–I) Percentage of CD11b⁺CD80⁺, CD11b⁺MHCII⁺, CD11c⁺CD80⁺, CD11c⁺MHCII⁺ cells in the spleen of infected mice. Data is representative of two independent experiments with five mice per group. Each graph represents mean ± SD (n = 5). ***$p < 0.0005$.

The online version of this article includes the following source data for figure 7:

**Source data 1.** AGK2 treatment enhances macrophage stimulation in the lungs and spleen of infected animals.

and IL17 producing CD4⁺ and CD8⁺ T cells in the lungs of AGK2-treated mice (*Figure 9A–E*). Interestingly, CD3⁺ cells isolated from the lungs of AGK2 treated and *Mtb*-infected mice displayed enhanced levels of NFκB p65 acetylation (*Figure 9F and G*). Upon SIRT2 inhibition, we observed a similar increase in the percentage of splenic CD4⁺ and CD8⁺ T cells producing pro-inflammatory cytokines IFNγ and IL17 (*Figure 9H–K*), indicating a shift in the immune balance toward Th1 and Th17 response. In summary, these data demonstrate the immunomodulatory effects of SIRT2 during *Mtb* infection.

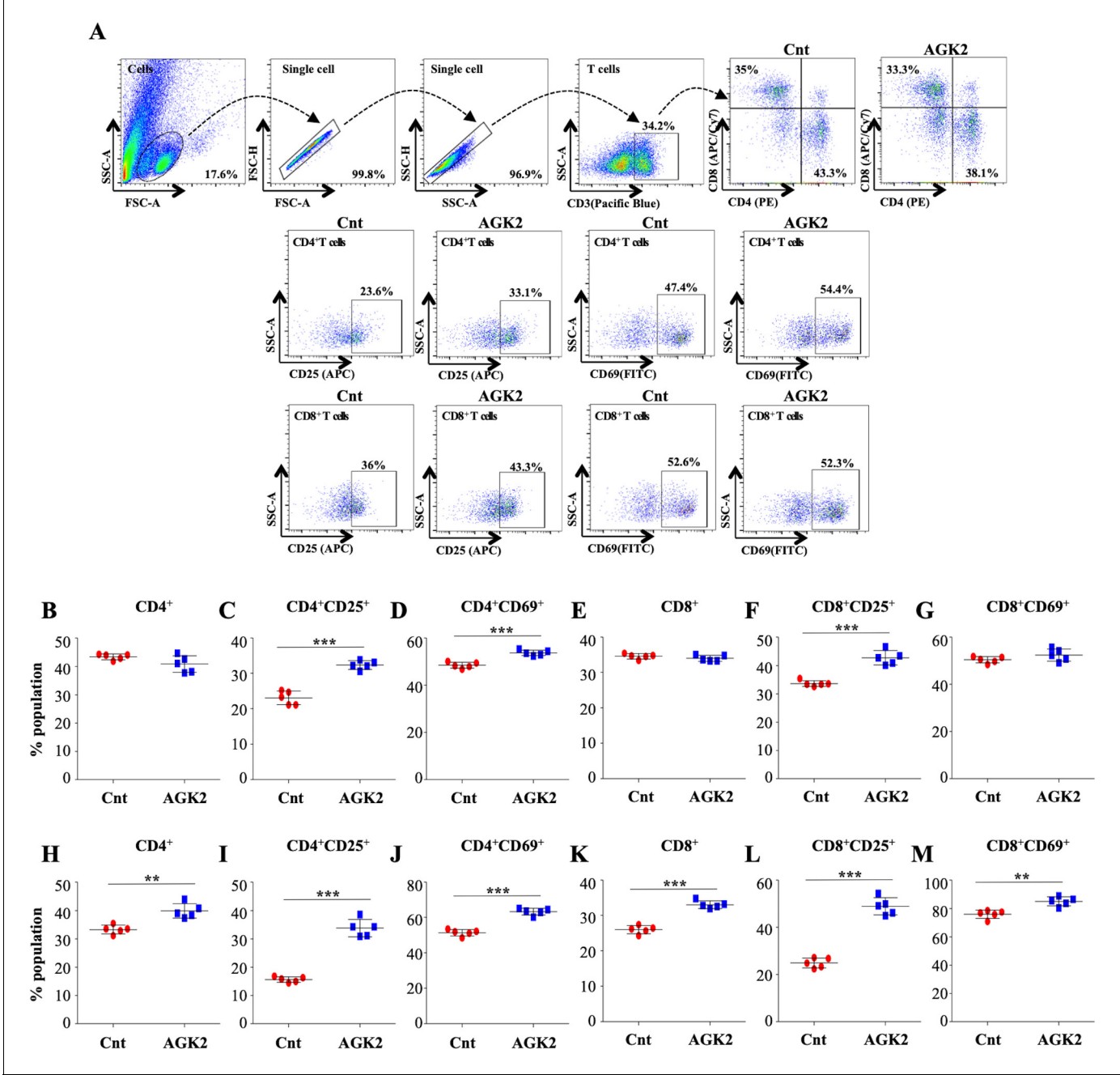

**Figure 8.** SIRT2 inhibition ameliorates T cell activation in the lungs and spleen of infected mice. Cells isolated from the lungs of control and AGK2-treated *Mtb*-infected C57BL/6 animals were subjected to surface staining (CD3-Pacific Blue, CD4-PE, CD8-APC/Cy7, CD69-FITC and CD25-APC) followed by FACS analysis. (**A**) Gating strategy and representative dot plots. Scatter plots depicting the percentage of CD4[+], CD4[+]CD69[+], CD4[+]CD25[+], CD8[+], CD8[+]CD69[+] and CD8[+]CD25[+] T cells in the (**B–G**) lungs and the (**H–M**) spleen of infected mice. The experiment was performed twice with five mice in each group. Data is represented as mean ± SD (n = 5). **p<0.005, ***p<0.0005.
The online version of this article includes the following source data for figure 8:

**Source data 1.** SIRT2 inhibition increases T cell activation in vivo.

## T cells from SIRT2-deficient mice transfer protective immunity against *Mtb*

To further demonstrate that protective immunity generated following AGK2 treatment was *Mtb* specific, we investigated whether adoptive transfer of T cells isolated from infected and AGK2-treated

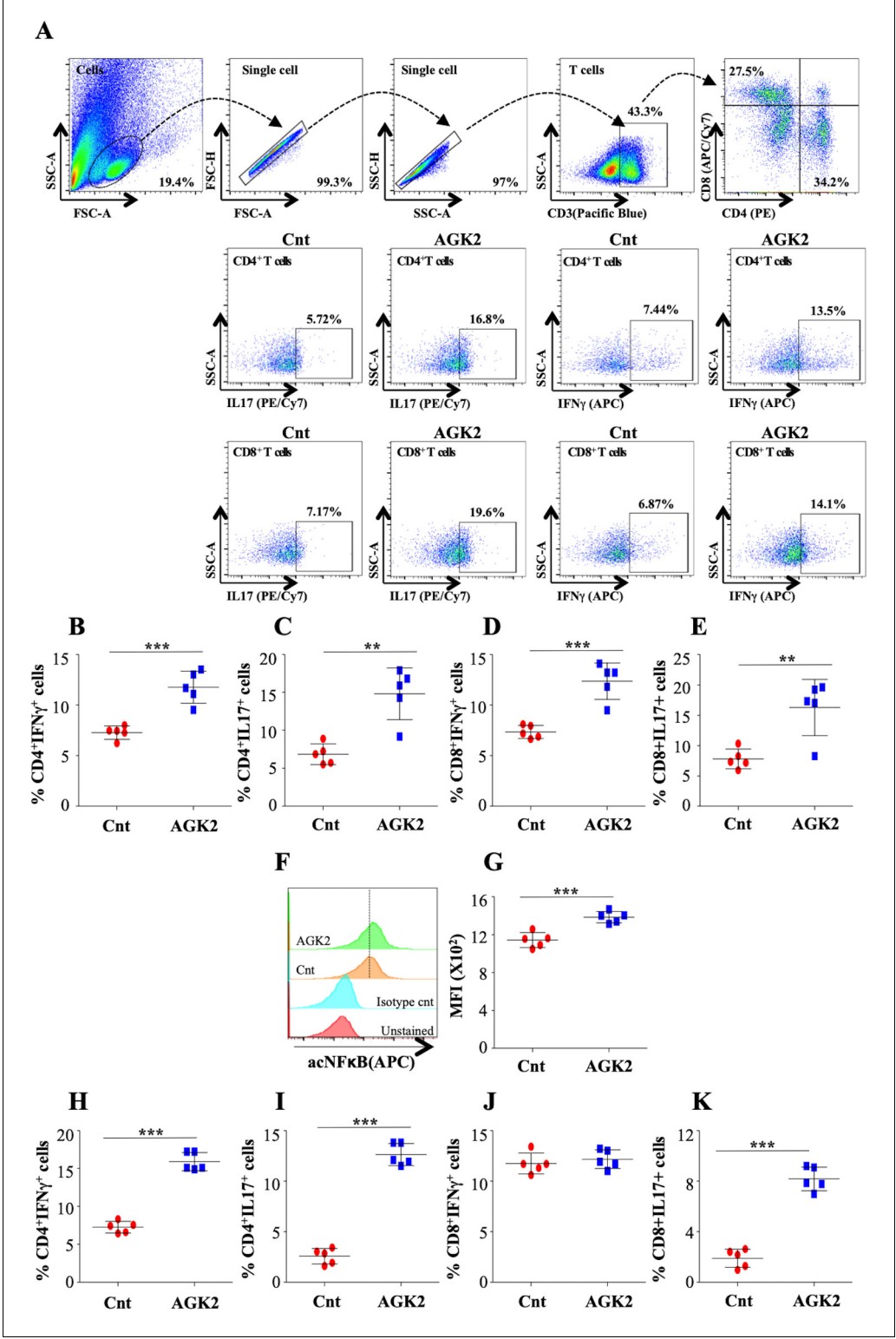

**Figure 9.** AGK2 treatment induces host protective immune responses against TB. Cells isolated from infected lungs of C57BL/6 mice were stained with CD3 (Pacific Blue), CD4 (PE), CD8 (APC/Cy7), IFNγ (APC) and IL17 (PE/cy7) followed by flow cytometry. (**A**) Gating strategy and representative flow cytometry dot plots. Percentage of IFNγ and IL17 producing CD4+ (**B and C**) and CD8+ (**D and E**) T cells in the lungs of infected mice. (**F**) Overlay plots and (**G**) quantification of acetylated NFκB p65 levels in the CD3+ T cells isolated from the lungs of infected mice. (**H–K**) IFNγ and IL17 expressing CD4+ (**H and I**) and CD8+ (**J and K**) T cells in the spleen of infected mice.
*Figure 9 continued on next page*

*Figure 9 continued*

The experiment was performed twice with five mice in each group. Data is represented as mean ± SD (n = 5). **p<0.005, ***p<0.0005.

The online version of this article includes the following source data for figure 9:

**Source data 1.** SIRT2 inhibition skews immune response toward Th1/Th17 phenotype Source data for *Figure 9B–D and G–K*.

mice confer resistance to *Mtb* infection in naive mice. For this, we intravenously transferred one million CD4$^+$ and CD8$^+$ T cells isolated from the spleen of from infected and AGK2-treated mice into *Rag1*$^{-/-}$ mice followed by low dose *Mtb* infection and CFU enumeration 21 days pi (*Figure 10A*). We first analyzed the overall health of these splenocytes by quantifying apoptosis and intracellular ROS. We observed that AGK2 treatment had no effect on the apoptotic levels in the splenic CD4$^+$ and CD8$^+$ T cells (*Figure 10B and C*). Similarly, ROS levels were also similar in the CD4$^+$ and CD8$^+$ T cells isolated from the spleen of infected and AGK2 treated animals (*Figure 10D and E*). To determine the bactericidal capacity of splenocytes, we performed co-culture experiments wherein GFP-H37Rv infected peritoneal macrophages were co-cultured with splenocytes isolated from the control or AGK2-treated infected mice for 48 hr followed by FACS analysis. Interestingly, mycobacterial growth was significantly restricted in macrophages co-cultured with T cells isolated from AGK2-treated mice (*Figure 10F*) indicating that AGK2 treatment enhances the efficacy of T cells in terms of stimulating macrophages for efficient bacterial clearance. Furthermore, T cells from AGK2-treated animals were more efficient in imparting protection to the *Rag1*$^{-/-}$ mice as compared to the T cells from *Mtb*-infected mice (Cnt) (*Figure 10G and H*). These data signify the effector functions of adoptively transferred T cells, in enhancing anti-mycobacterial activity.

## AGK2 as an adjunct therapy against susceptible and drug-resistant tuberculosis

In terms of TB pathology, both the C57BL/6 and BALB/c exhibit similar resistant phenotype to infections (*Keller et al., 2006*). In order to reinforce the protective effects of AGK2, we repeated in vivo infections in BALB/c mice (*Figure 11A*). AGK2 treatment significantly lowered the bacterial load in the lungs (*Figure 11B*), spleen (*Figure 11C*) and liver (*Figure 11D*) of infected animals as compared to the control group. Lastly, to examine the efficacy of AGK2 as an adjunct to anti-TB therapy we designed a combinatorial therapy wherein infected mice were treated with a combination of AGK2 and front line anti-TB drug INH for 15 days followed by CFU enumeration (*Figure 11E*). Co-treatment of AGK2 and INH significantly decreased the bacterial burden in the lungs as compared to the mice treated with either AGK2 or INH alone (*Figure 11F*). Interestingly, AGK2 treatment significantly decreased the bacillary load of MDR as well as XDR strains of TB (*Figure 11G–I*). One of the advantages of host-directed therapies (HDTs) is the reduction in the risk of generation of drug resistance. HDTs provide untapped opportunities to eliminate drug-resistant strains by manipulating host immune system.

## Discussion

To combat host-induced stresses, *Mtb* has evolved several survival strategies. Recent studies hint toward the pathogen-induced histone modifications as an emerging host subversion strategy employed by a number of bacteria and viruses for successful persistence inside the host. Virulence factors and toxins released by the pathogens have been shown to alter host epigenome by modulating the activity of HATs and HDACs. *Mtb* was shown to induce histone deacetylation at the HLA-DRα promoter leading to the inhibition of IFN-γ-dependent HLA-DR gene expression (*Wang et al., 2005*). Since this study, many pathogens were shown to influence host histone acetylome (*Grabiec and Potempa, 2018*). For instance, virulence factor listeriolysin-O (LLO) of *L. monocytogenes* causes global histone H4 deacetylation, whereas invasin protein InlB mediates histone H3K18 deacetylation both of which lead to transcriptional reprograming of the host (*Eskandarian et al., 2013*; *Hamon et al., 2007*). Upregulation of HDACs by intracellular pathogens like *Anaplasma phagocytophilum*, *Pseudomonas aeruginosa*, *Porphyromonas gingivalis* led to epigenetic suppression of host defense genes. Lately, pathogens were shown to target sirtuins for subverting host acetylome.

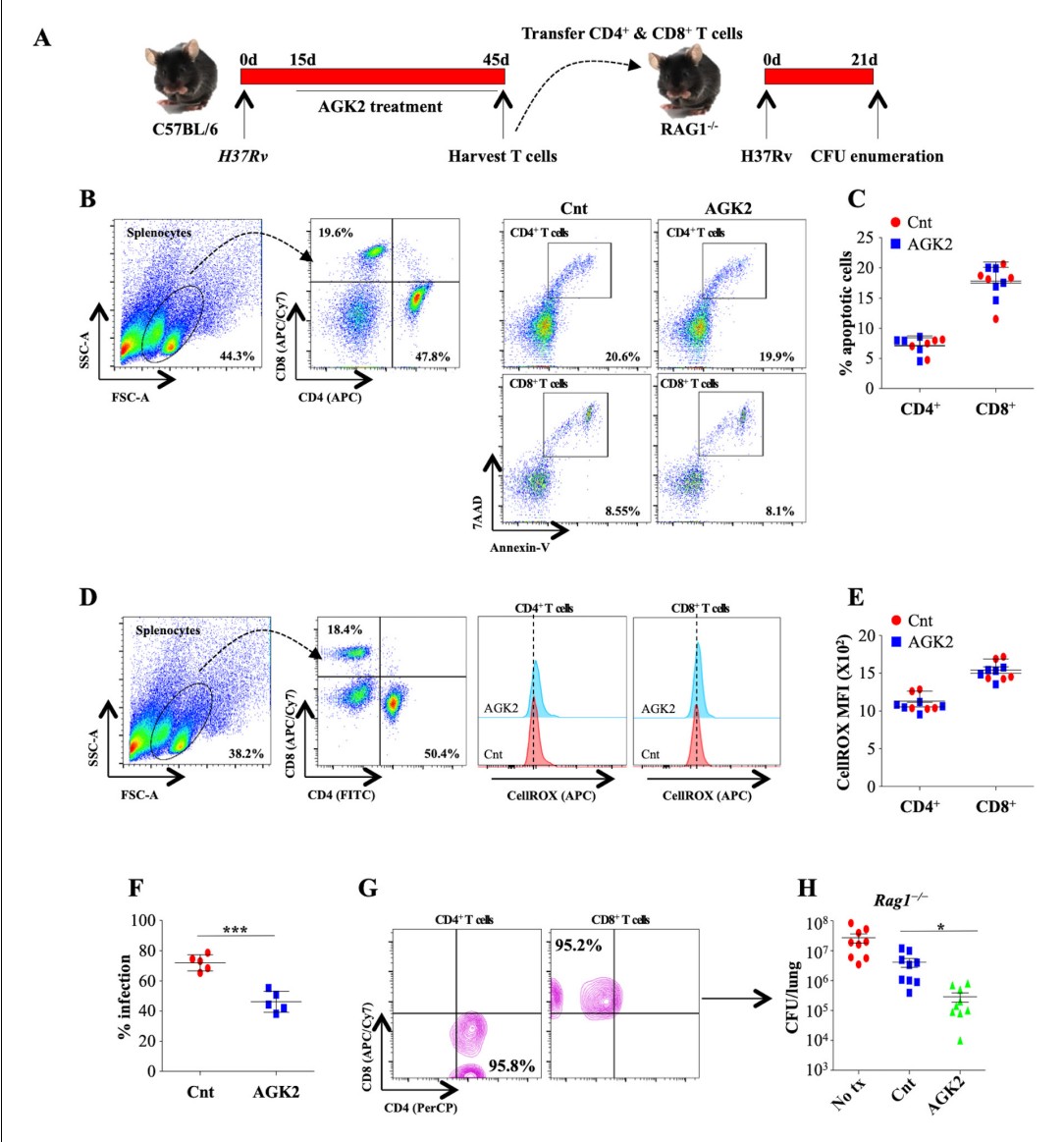

**Figure 10.** AGK2-treated splenocytes demonstrate increased propensity toward mycobacterial killing and transfer *Mtb*-specific protective immunity in naive animals. (A) Schematic representation of the adoptive transfer experiment. Splenocytes isolated from infected and AGK2-treated C57BL/6 mice were analyzed for (B and C) apoptosis by Annexin V staining and (D and E) ROS production via cellROX (see Materials and methods). (F) Splenocytes isolated from infected and AGK2 treated mice were co-cultured with peritoneal macrophages infected with GFP expressing H37Rv. 48 hr pi, percentage of infected cells were analyzed by flow cytometry. CD4+ and CD8+ T cells purified from the spleen of infected and AGK2-treated mice (G) were intravenously transferred into *Rag1-/-* mice followed by challenge with aerosolized *Mtb*. (H) CFU enumeration 21 days after adoptive transfer. Data is representative of two independent experiments performed with four to five mice in each group. Data is represented as mean ± SD (n = 5). H shows combined data from two independent experiments (n = 9). *p<0.05, ***p<0.0005.

The online version of this article includes the following source data for figure 10:

**Source data 1.** AGK2-treated splenocytes transfer *Mtb*-specific protective immunity in naive animals.

*Salmonella typhimurium* degrades SIRT1 to evade autophagy (*Ganesan et al., 2017*) while it upregulates SIRT2 to regulate NOS2 activity (*Gogoi et al., 2018*). Recently, it was shown that activation of SIRT1 aids *Mtb* survival by inducing phagosome-lysosome fusion and autophagy (*Cheng et al., 2017*). In this study, we for the first time show significant upregulation of SIRT2 during *Mtb* infection in peritoneal macrophages. Inhibition of SIRT2 activity by AGK2 as well as a decrease in the SIRT2 levels following siRNA treatment led to decreased intracellular survival of *Mtb*.

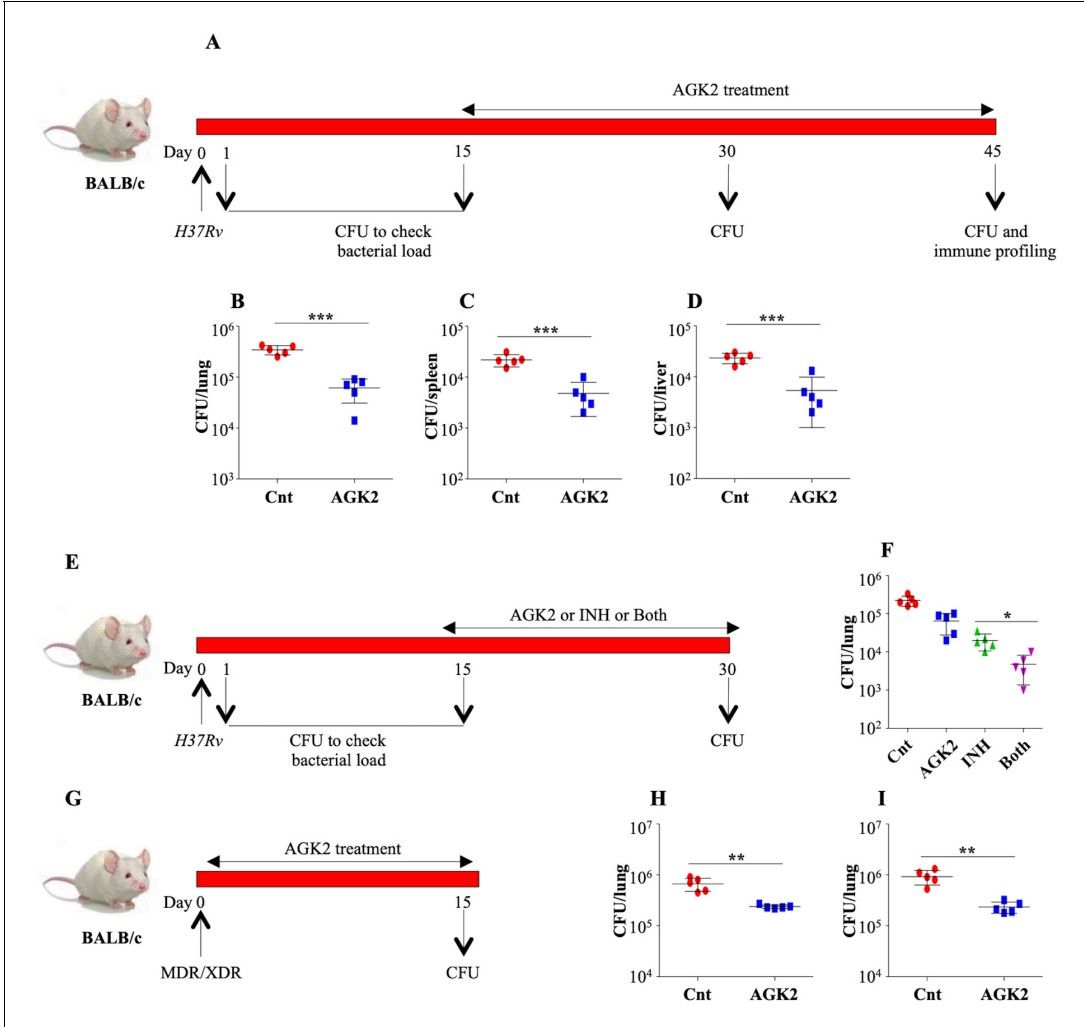

**Figure 11.** AGK2 treatment increases the efficacy of anti-TB drug INH. (**A**) Experimental plan for infection in BALB/c mice. (**B–D**) CFU enumeration to determine the bacterial burden in the lungs, spleen and liver of infected mice. (**E**) Schematic representation of the adjunct therapy experiment. A group of mice were infected with low dose of *H37Rv*. Following a rest of 15 days, mice were either left untreated or were treated with AGK2, INH or both. After 15 days of treatment, mice were euthanized for CFU enumeration in the lungs. (**F**) CFU from the lung homogenates of respective animals. (**G**) Schematic representation of the AGK2 treatment infection model. (**H and I**) CFU obtained from the lung homogenates of mice infected with MDR and XDR strains of *Mtb* with and without AGK2 treatment. The experiment was performed once with five mice in each group. Data is represented as mean ± SD (n = 5). *p<0.05, **p<0.005, ***p<0.0005.

The online version of this article includes the following source data for figure 11:

**Source data 1.** Efficacy of AGK2 as an adjunct to current anti-TB drug INH.

APCs play a critical role in TB pathogenesis by directly killing the bacteria and by activating T cells and thus shaping the extent and nature of the adaptive immune response. Expression and engagement of MHC and costimulatory molecules on the surface of APCs activates these cells as well as T cells leading to the release of pro-inflammatory cytokines such as IFNγ and IL17 which are shown to play a protective role in TB disease. IFNγ deficiency in humans and mice is associated with increased susceptibility to *Mtb* (*Flynn et al., 1993*; *Zhang et al., 2008*). Apart from being a protective cytokine, IL17 is also essential for recall responses to *Mtb*(*Chatterjee et al., 2011*). SIRT2 has been shown to inhibit the stress-induced expression of pro-inflammatory cytokines and surface activation markers in different cell lines (*Kim et al., 2013*; *Lo Sasso et al., 2014*; *Pais et al., 2013*). In our study, AGK2 treatment increased the expression of activation markers on the surface of mouse peritoneal macrophages upon *Mtb* infection leading to a prominent T cell activation and elevated levels of proinflammatory cytokines. A similar effect of SIRT2 inhibition was observed in *Mtb* specific T cells

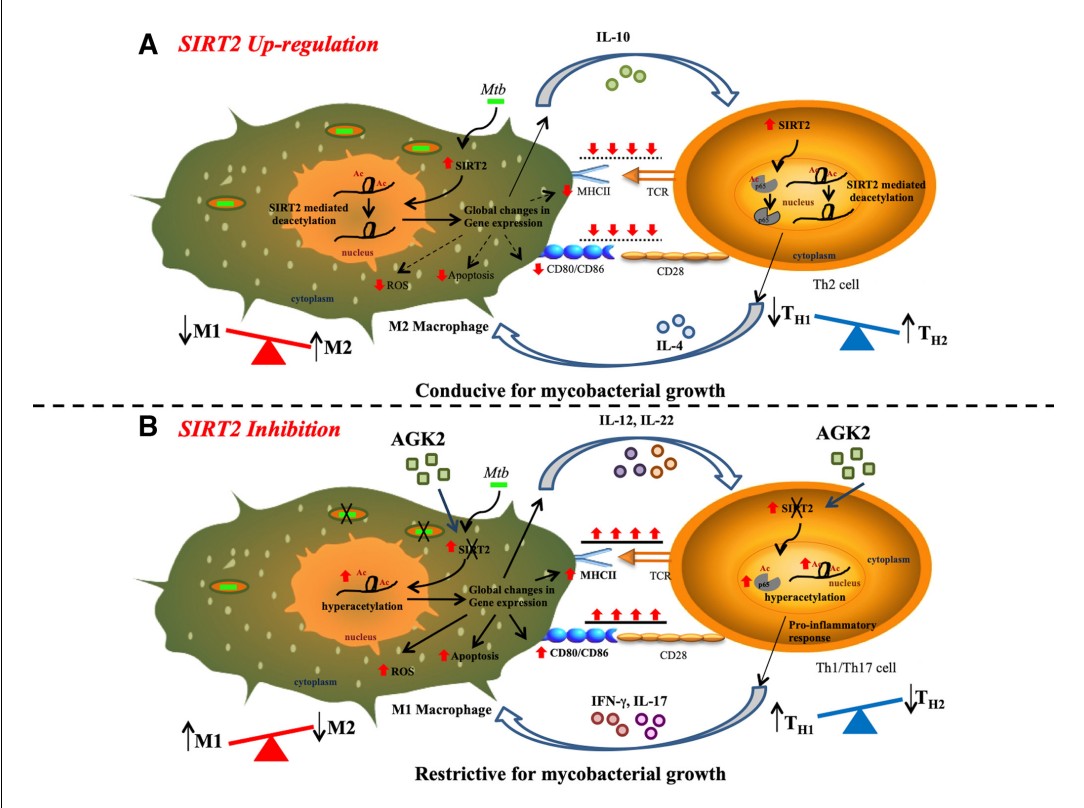

**Figure 12.** Proposed mechanism of SIRT2 up-regulation and consequence of SIRT2 inhibition on *Mtb* survival. (**A**) *Mtb* infection induces the expression and translocation of SIRT2 in the nucleus where it deacetylates histone H3K18 to induce consequential changes in host transcriptome leading to decreased activation of macrophages, reduced levels of apoptosis and ROS, and increased expression of anti-inflammatory Th2-polarizing cytokines. Simultaneously, upregulation of SIRT2 in *Mtb*-specific T-cells specifically deacetylates and deactivates NFκB p65 leading to an anti-inflammatory response. M2 macrophages and Th2 cells preferentially generated provide a more favorable environment for *Mtb* survival within the host. (**B**) SIRT2 inhibition in *Mtb*-infected cells reverses the SIRT2-dependent gene expression changes enhancing the capacity of macrophages to activate T-cells. AGK2 treatment also enhances anti-mycobacterial defense forces in macrophages such as apoptosis, ROS and pro-inflammatory Th1/Th17 polarizing cytokines. Moreover, hyperacetylation of NFκB p65 in *Mtb*-specific activated T-cells leads to Th1/Th17-mediated pro-inflammatory immune response. Enhanced levels of protective innate and adaptive immunity in AGK2 treated cells restrict mycobacterial survival.

wherein AGK2 treatment led to enhanced acetylation of histone H3K18 and NFκB p65 leading to the induction of proinflammatory genes (*Cheng et al., 2017*). For the first time, we demonstrate that SIRT2 inhibition in T cells shifts the type-1/type-2 balance toward protective Th1/Th17 response during TB pathogenesis.

The protective role of SIRT2 deficiency has been well documented in age-related neurodegenerative diseases (*Fourcade et al., 2018*). However, very few studies talk about SIRT2 dependent disease progression in bacterial and viral infections (*Ciarlo et al., 2017*; *Gogoi et al., 2018*; *Piracha et al., 2018*). As a well-established SIRT2 inhibitor, several studies have used AGK2 treatment in mice models with no visible side effects (*Wang et al., 2016a*). Consequently, we extended our study in the murine model of TB. Mice administered with AGK2 displayed a significant reduction in bacterial load, which translated into diminished tissue pathology evident by the negligible presence of granulomatous lesions in the lungs. Innate cytokine milieu induced by SIRT2 inhibition drive the adaptive immune response toward Th1 and Th17 cells. Cytokine profiling indicated a robust pro-inflammatory response in the lungs and the spleen of AGK2 treated and *Mtb*-infected mice. SIRT2 has been shown to regulate immuno-metabolism in sepsis (*Wang et al., 2018*). SIRT2 inhibits inflammatory response in obese mice via deacetylating NFKB p65 (*Wang et al., 2016b*). Treatment of SIRT2 specific inhibitor AK7 enhanced LPS-induced pro-inflammatory cytokine response in RAW 264.7 macrophages

(*Wang et al., 2016b*). *Mtb* upregulates the expression of IL-10 via TLR2-ERK pathway to promote TB pathogenesis (*Beamer et al., 2008*; *Richardson et al., 2015*). Interestingly, AGK2 treatment decreases the phosphorylation of ERK1/2 in peritoneal macrophages. Furthermore, previous reports suggest that high SIRT2 expression is associated with upregulation of IL-10 (*Deng et al., 2016*; *Lo Sasso et al., 2014*). The immune response generated during AGK2 treatment was *Mtb* specific as was evident by adoptive transfer studies wherein T cells from AGK2 treated and *Mtb* infected mice when transferred into naive *Rag1*$^{-/-}$ mice conferred resistance to *Mtb* infection. Moreover, splenocytes from mice treated with AGK2 displayed a higher ability to promote *Mtb* killing by macrophages. In mycobacterial infections, apoptosis of infected macrophages is thought to prevent bacterial spread and restrict mycobacterial growth by inducing host immune response (*Srinivasan et al., 2014*). However, apoptosis of T cells results in T cell exhaustion and is correlated with defective immune response. Noteworthy is the fact that SIRT2 deficiency resulted in increased apoptosis in peritoneal macrophages with no effect on T cells which is in agreement with earlier reports demonstrating the anti-apoptotic effects of SIRT2 in macrophages (*Li et al., 2011*; *Zhang et al., 2018*).

SIRT2 is mainly considered to be active in the cytosol where it targets many cellular proteins to regulate multiple pathways such as microtubule dynamics (α-tubulin, [*North et al., 2003*], adipogenesis (FOXO1, [*Jing et al., 2007*], mitochondrial biogenesis (PGC-1α, [*Krishnan et al., 2012*], metabolism (ACLY, PEPCK1, G6PD, [*Jiang et al., 2011*; *Lin et al., 2013*; *Wang et al., 2014*], ROS suppression (FOXO3a, [*Wang et al., 2007*], inflammation (NF-κB, [*Rothgiesser et al., 2010*] and many more [*Gomes et al., 2015*]. However, SIRT2 has also been shown to transiently shuttle to the nucleus under certain conditions where it targets both histone and non-histone proteins. It regulates mitosis by deacetylating H4K16 and DNA-damage induced hyper-acetylation by targeting P300 and H3K56 (*Houtkooper et al., 2012*). Recently, Eskandarian et al uncovered a novel nuclear function of SIRT2 where it regulates host gene expression by deacetylating H3K18 during *Listeria* infection. Our data also suggests nuclear migration of SIRT2 upon *Mtb* infection where it deacetylates H3K18 to influence gene expression. In line with the above observations, our RNAseq data revealed a similar trend where AGK2 treatment did not influence the gene expression pattern of uninfected macrophages, whereas out of 992 total genes affected by *Mtb* infection, the expression of more than 50% (578) of the genes was dependent on SIRT2 activity. KEGG pathway analysis performed on SIRT2-dependent genes revealed a significant number of pathways, which have a known role to play in the containment of *Mtb* infection. For instance, there was a significant up-regulation in genes involved in antigen processing and presentation, Th17 cell differentiation, cytokine-cytokine receptor interactions and TNF signaling.

*Mtb* infection results in the phosphorylation of PI3K and Akt while inhibiting the activity of either PI3K or Akt leads to the suppression of mycobacterial growth in the host cell via upregulation of matrix metalloproteinases and multiple pro-inflammatory cytokines (*Brace et al., 2017*; *Huang et al., 2012*). Interestingly, PI3K-Akt signaling and TLR signaling pathways were downregulated upon AGK2 treatment.

Although antimicrobial effects of SIRT2 inhibition have been demonstrated in *Salmonella* and *Listeria*, this is the first report suggesting modulation of host adaptive immune responses as one of the mechanisms by which AGK2 inhibits the growth of an intracellular pathogen. These data are further strengthened by our transcriptome studies which provide an insight into the mechanistic details of how SIRT2 activity aids in mycobacterial survival. The proposed mechanism of mycobacterial restriction via AGK2 is depicted in *Figure 12*.

With this background we attempted to put forth SIRT2 inhibitors as potential candidates for HDTs in adjunct with the existing anti-TB regime and to our satisfaction, AGK2 was successful in further reducing the bacillary load in INH-treated infected mice. However, further studies on different experimental models such as non-human primates and with a series of other chemical inhibitors of SIRT2 are required to fully explore the usability of SIRT2 inhibitors as potential TB HTDs. Nevertheless, the ability of AGK2 to effectively target drug-resistant strains of *Mtb* is an encouraging finding towards the development of newer TB therapeutics.

# Materials and methods

## Key resources table

| Reagent type (species) or resource | Designation | Source or reference | Identifiers | Additional information |
|---|---|---|---|---|
| Genetic reagent (*M. musculus*) | B6.129S7-Rag1$^{tm1Mom}$/J | PMID:1547488 | RRID:IMSR_JAX:002216 | |
| Cell line (*M. musculus*) | RAW 264.7 | ATCC | RRID:CVCL_0493 | |
| Antibody | Rabbit monoclonal [EP959Y] to Histone H3 (acetyl K18) | Abcam | Cat#ab40888 | 1:10000 (WB) 1:2000 (FC) |
| Antibody | Rabbit polyclonal to NF-kB p65 (acetyl K310) | Abcam | Cat#ab19870 | 1:5000 (WB) 1:1000 (FC) |
| Antibody | Rabbit polyclonal to beta Actin | Abcam | Cat#ab8227 | 1:5000 (WB) |
| Antibody | Mouse monoclonal [GT114] to alpha Tubulin | Abcam | Cat#ab184613 | 1:5000 (WB) |
| Antibody | Rabbit monoclonal [EPR16772] to alpha Tubulin (acetyl K40) | Abcam | Cat#ab179484 | 1:2500 (WB) 1:1000 (FC) |
| Antibody | Rabbit monoclonal [EPR20411] to SIRT2 | Abcam | Cat#ab211033 | 1:2500 (WB) 1:1000 (FC) |
| Antibody | FITC anti-mouse CD4 Antibody | Biolegend | Cat#100510 | 1:250 (FC) |
| Antibody | PerCP anti-mouse CD4 Antibody | Biolegend | Cat#100433 | 1:250 (FC) |
| Antibody | APC anti-mouse CD4 Antibody | Biolegend | Cat#100411 | 1:250 (FC) |
| Antibody | APC/Cyanine7 anti-mouse CD8a Antibody | Biolegend | Cat#100713 | 1:250 (FC) |
| Antibody | FITC anti-mouse CD69 Antibody | Biolegend | Cat#104506 | 1:250 (FC) |
| Antibody | APC anti-mouse CD25 Antibody | Biolegend | Cat#101910 | 1:250 (FC) |
| Antibody | APC/Cy7 anti-mouse/human CD11b Antibody | Biolegend | Cat#101226 | 1:250 (FC) |
| Antibody | APC anti-mouse CD11c Antibody | Biolegend | Cat#117310 | 1:250 (FC) |
| Antibody | APC anti-mouse IFN-γ Antibody | Biolegend | Cat#505810 | 1:250 (FC) |
| Antibody | PE/Cyanine7 anti-mouse IL-17A Antibody | Biolegend | Cat#506922 | 1:250 (FC) |
| Antibody | PE anti-mouse CD4 Antibody | Biolegend | Cat#100512 | 1:250 (FC) |
| Antibody | FITC anti-mouse CD80 Antibody | Biolegend | Cat#104706 | 1:250 (FC) |
| Antibody | PE anti-mouse I-A/I-E Antibody | Biolegend | Cat#107607 | 1:250 (FC) |
| Antibody | Pacific Blue anti-mouse CD3 Antibody | Biolegend | Cat#100214 | 1:250 (FC) |
| Antibody | p44/42 MAPK (Erk1/2) Antibody | Cell Signaling Technology | Cat#9102 | 1:2000 (WB) |

*Continued on next page*

*Continued*

| Reagent type (species) or resource | Designation | Source or reference | Identifiers | Additional information |
|---|---|---|---|---|
| Antibody | Phospho-p44/42 MAPK (Erk1/2) (Thr202/ Tyr204) Antibody | Cell Signaling Technology | Cat#9101 | 1:1000 (WB) |
| Antibody | p38 MAPK Antibody | Cell Signaling Technology | Cat#9212 | 1:2000 (WB) |
| Antibody | Phospho-p38 MAPK (Thr180/Tyr182) (D3F9) XP Rabbit mAb | Cell Signaling Technology | Cat#4511 | 1:1000 (WB) |

## Mice

BALB/c, C57BL/6 and *Rag1*[-/-](6–8 weeks of age) female mice were procured from NII, New Delhi, India. All animals were maintained in the animal facility of NII, New Delhi, India.

## Bacteria

The mycobacterial strains used in this study (H37Rv, MDR and XDR) were maintained in 7H9 (Middlebrook, Difco) medium supplemented with 10% ADC (albumin, dextrose, and catalase; Difco), 0.05% Tween 80% and 0.2% glycerol. Cultures grown to mid-log phase were preserved in 20% glycerol at −80˚C. Infections were carried out from these cryopreserved stocks.

## Peritoneal macrophages

Two ml of 4% thioglycollate (Sigma) was injected into C57BL/6 mice intraperitoneally. Five days later, peritoneal cavity was washed with ice cold PBS in order to extract peritoneal exudates cells. These cells were cultured in RPMI-1640 medium supplemented with 10% fetal bovine serum (Thermo fisher scientific Inc or Hyclone). After overnight incubation at 37˚C and 5% $CO_2$, cells were washed with PBS to remove non-adherent cells. Adherent monolayer cells were confirmed to be pure macrophages by staining with CD11b antibody followed by flow cytometry (FACS Verse Cell-Sorting System, BD Biosciences). FACS Data was analyzed using the FACSuite software (BD Biosciences).

## AGK2 treatment

The peritoneal macrophages were pre-treated with 10 μM of AGK2 for 2 hr following infection with *Mtb* at an MOI of 10. 4 hr after infection, the cells were washed twice with 1X PBS to remove the extracellular bacilli. The infected cells were then maintained in RPMI-1640 medium supplemented with 10% fetal bovine serum along with 10 μM of AGK2 throughout the course of the experiment.

Transfection shRNA transfections were carried out in RAW 264.7 macrophages (no mycoplasma contamination) seeded in 12-well plates at the density of $0.5 \times 10^6$ cells per well using Lipofectamine 2000 Transfection Reagent (Invitrogen) as per manufacturer's instructions (*Bhaskar et al., 2011*). 24 hr post transfection, cells were infected with GFP expressing H37Rv for 4 hr. After washing twice with PBS to remove extracellular bacilli, cells were incubated in a 5% $CO_2$ incubator at 37˚C for 48 hr followed by flow cytometry in order to detect intracellular bacterial fluorescence (MFI) and percentage of infected cells. Cell lysates were also plated onto 7H11 plates for CFU enumeration.

## Bacterial infections and co-culture of macrophages with T cells

Peritoneal macrophages were cultured at the density of $1 \times 10^6$ cells per ml of media. Cells were infected with *Mtb* H37Rv (MOI 10). For infection, cryostocks of bacteria were resuspended in 1 ml PBS and passed through 26-gauge needles five times in order to achieve single-cell suspensions. For co-culture experiments, the spleen from infected animals were harvested and single-cell suspensions were made using frosted glass slides followed by RBC lysis. Cells were counted and seeded with the macrophages in a ratio of 1:10. After 48 hr, macrophages and splenocytes were screened for different surface markers and intracellular cytokines by flow cytometry.

## Fluorescent microscopy

Peritoneal macrophages were plated on cover slips and infected with H37Rv-GFP at an MOI of 10. After 4hof infection, cells were fixed with 4% paraformaldehyde and permeabilized with 0.2% (w/v) Triton X-100 in PBS for 20 min. This was followed by primary (SIRT2, Abcam) and secondary (alexa594, CST) antibody staining as per the manufacturer's instructions. After washing thoroughly with PBS, the cover slips were mounted onto glass slides with DAPI containing Vectashield mountant (H-1200, Vector Labs). Several fields were acquired randomly from each set with a Carl Zeiss epi-fluoresence microscope. Images were saved as 16-bit TIF files and analyzed by ImageJ software (http://rsb.info.nih.gov/ij/).

## Western blot analysis

RIPA buffer (50 mM Tris,pH 8.0, 150 mM NaCl, 1.0% NP-40, 0.5% Sodium deoxycholate, 0.1% SDS) freshly supplemented with PhosSTOP and complete protease inhibitor procured from Roche was used to prepare whole cell lysates of peritoneal macrophages. Samples were electrophoresed on SDS-PAGE and electroblotted onto nitrocellulose membranes (Bio-Rad). After blocking with 5% BSA prepared in PBST (PBS and 0.05% Tween-20), blots were probed for different proteins using corresponding antibodies acquired from Abcam and CST. Blots were developed on autoradiograms using chemiluminescent HRP substrate (ECL, Millipore). Histones were extracted using histone extraction kit from Abcam, USA as per manufacturer's details. Cell lysates were fractionated into cytosolic and nuclear fractions using nuclear extraction kit from Abcam, USA according to the protocol provided by the manufacturer.

## qPCR analysis

Total RNA was isolated from peritoneal macrophages either uninfected or infected with H37Rv with and without AGK2 treatment using RNAeasy isolation kit (QIAGEN, Germany) according to the manufacturer's instructions. cDNA was synthesized using iscript cDNA synthesis kit (Bio-Rad) followed by qRT-PCR using SYBR Green Master Mix (Bio-Rad). Real-time quantitative RT-PCR analysis was performed using BioRad Real-Time thermal cycler (BioRad, USA). The primers used in the study are:

| Primer | Sequence (5'−3') |
|---|---|
| *H2-Aa* Forward Primer | TCAGTCGCAGACGGTGTTTAT |
| *H2-Aa* Reverse Primer | GGGGGCTGGAATCTCAGGT |
| *CD74* Forward Primer | CCGCCTAGACAAGCTGACC |
| *CD74* Reverse Primer | ACAGGTTTGGCAGATTTCGGA |
| *IL22* Forward Primer | ATGAGTTTTTCCCTTATGGGGAC |
| *IL22* Reverse Primer | GCTGGAAGTTGGACACCTCAA |
| *CD14* Forward Primer | CTCTGTCCTTAAAGCGGCTTAC |
| *CD14* Reverse Primer | GTTGCGGAGGTTCAAGATGTT |
| *CCL7* Forward Primer | GCTGCTTTCAGCATCCAAGTG |
| *CCL7* Reverse Primer | CCAGGGACACCGACTACTG |
| *CXCL1* Forward Primer | CAAGGCTGGTCCATGCTCC |
| CCL7 Reverse Primer | TGCTATCACTTCCTTTCTGTTGC |
| GNG7 Forward Primer | TCAGGTACTAACAACGTCGCC |
| GNG7 Reverse Primer | CAGTAGCCCATCAGGTCTGAC |
| *ITGA3* Forward Primer | CCTCTTCGGCTACTCGGTC |
| GNG7 Reverse Primer | CCGGTTGGTATAGTCATCACCC |
| *STAT1* Forward Primer | TCACAGTGGTTCGAGCTTCAG |
| *STAT1* Reverse Primer | GCAAACGAGACATCATAGGCA |
| *CREB5* Forward Primer | AGGATCTTCTGCCGTCTTGAT |

*Continued on next page*

*Continued*

| Primer | Sequence (5'−3') |
|--------|------------------|
| *CREB5* Reverse Primer | GCGCAGCCTTCAGTCTCAT |
| *IL6* Forward Primer | TGGGGCTCTTCAAAAGCTCC |
| *IL6* Reverse Primer | AGGAACTATCACCGGATCTTCAA |
| *ZBP1* Forward Primer | AAGAGTCCCCTGCGATTATTTG |
| *ZBP1* Reverse Primer | TCTGGATGGCGTTTGAATTGG |
| *SIRT2* Forward Primer | GAGGTGGCATGGATTTTGAC |
| *SIRT2* Reverse Primer | AGATGGTAGTGCTGGGGTTG |
| *ACTB* Forward Primer | AGTGTGACGTTGACATCCGTAAAGA |
| *ACTB* Reverse Primer | GGACAGTGAGGCCAGGATGG |

## RNAseq analysis

Mouse peritoneal macrophages were infected with *Mtb* at an MOI of 10 and harvested 24 hr pi for RNA isolation. RNAseq was carried out by Macrogen, Inc in collaboration with Bencos Research Solutions Pvt. Ltd. for data analysis. Briefly, the methodology included generation of raw sequencing reads followed by adapter trimming and QC filtering of reads:Phred score (Q) > 30, alignment to reference genome (mapped unique reads >80%), normalization and read count generation and finally differential expression determined using DESeq2 (significant differentially expressed genes selected based on FDR < 0.05). The quality assessment was carried out on raw Fastq files received using NGS QC toolkit with a minimum average quality (Q) value of 20 and an overall cutoff of 70%. Filtered reads were aligned with STAR alignment tool. GRCm38 was used as a reference genome. HTSeq was used to quantify the reads for all genes. After quantification, DESeq2 was used for differential gene expression analysis. GO enrichment was carried out using the R bioconductor package 'ClusterProfiler'.

## *Mtb* infection of mice and enumeration of colony forming units (CFU)

Mice were infected with *Mtb* via the aerosol route using a Madison aerosol chamber (University of Wisconsin, Madison, WI) with its nebulizer pre-calibrated to deposit a total of ~110 to the lungs of each mouse as previously described (*Bhattacharya et al., 2014*). Briefly, mycobacterial stocks recovered from a −80°C freezer were quickly thawed and passed through 26 gauge syringe five times to obtain a single-cell suspension. Fifteen ml of the bacterial cell suspension ($1.5 \times 10^6$ cells per ml) was placed in the nebulizer of the Madison aerosol chamber. Five randomly selected mice were sacrificed at various time points and organs were harvested, homogenized in 0.2 µm filtered PBS containing 0.05% Tween 80, plated onto 7H11 Middlebrooks (Difco USA) plates containing 10% oleic acid, albumin, dextrose and catalase (OADC) (Difco, USA) and incubated at 37°C for 21–28 days. Bacterial colonies were counted and CFU were estimated as per dilution. Half the spleen and the lung from each mouse was used for immune cell profiling.

## Drug administration

For ex vivo experiments, peritoneal macrophages were treated with 10 µM AGK2. 20 mg/kg of SIRT-2 inhibitor (AGK2, Abcam) in 100 µl of PBS containing 5% DMSO v/v was administered intraperitoneally to the mice thrice a week for 45 days, whereas control mice were given vehicle only. INH (100 mg/L) was constantly present in the drinking water of treated animals. The water was changed every second or third day.

## Flow cytometry: surface and intracellular staining

Ex vivo: Splenocytes and macrophages from co-culture experiments were taken out and stained for different surface markers and intracellular cytokines. Briefly, macrophages were surface stained with CD11b (APC/Cy7), CD11c (APC), CD80 (FITC) and MHC-II (PE). Splenocytes were surface stained with CD3 (Pacific Blue), CD4 (PE), CD8 (APC/Cy7), CD69 (FITC) and CD25 (APC). For intracellular

cytokine analysis, splenocytes were stained with CD3 (Pacific Blue), CD4 (PE), CD8 (APC/Cy7), IFNγ (APC) and IL17 (PE/cy7).

In vivo: The lungs and the spleen were isolated from mice and macerated by frosted slides in ice cold RPMI 1640 containing 10% FBS to prepare single-cell suspension. Red blood cells (RBCs) were lysed with RBC cell lysis buffer and cells were washed with 10% RPMI 1640. $1 \times 10^6$ cells per surface marker were used for surface staining. For intracellular staining, $1 \times 10^6$ cells were cultured per well in 12-well plates, activated with H37Rv Complete Soluble Antigen o/n, followed by treatment with 10 μg/ml Brefeldin A (eBiosciences, USA) for 6 hr. Cells were washed twice with PBS and stained with fluorescently labeled antibodies directed against surface markers. After staining, cells were washed again with PBS and fixed with 100 μl fixation buffer (eBiosciences, USA) for 30 min, then re-suspended in 200 μl permeabilization buffer (eBiosciences, USA) and stained with fluorescently labeled anti-cytokine antibodies. For detecting the levels of SIRT2, histone H3K18ac, acetylated NFκB p65 and acetylated α-tubulin, permeabilized cells were stained with appropriate primary anti-bodies followed by staining with secondary antibody labeled with Alexa Fluor 647 or isotype control IgG-APC. The intensity of fluorochrome-labeled cells was measured by flow cytometry (FACS Verse Cell-Sorting System, BD Biosciences). Data analysis was performed by FlowJo (Tree star, USA).

## T cell adoptive transfer

The spleen from infected and AGK2 treated animals were harvested and single-cell suspensions were made using frosted glass slides followed by RBC lysis. Cells stained with the anti-CD4 and anti-CD8 antibodies were sorted into CD4$^+$ and CD8$^+$ pure population in FACS Aria (BD Biosciences) and cultured overnight in complete RPMI medium. One million of pooled CD4$^+$ and CD8$^+$ T cells were transferred intravenously into *Rag1*$^{-/-}$ mouse. Two days later, the mice were infected with low dose infection with aerosolized *Mtb*.

## Apoptosis and cellular ROS

Peritoneal macrophages or splenocytes isolated from infected animals were analyzed for apoptosis with the help of FITC annexin V apoptosis detection kit with 7-AAD from Biolegend as per the manu-facturer's protocol. Cellular ROS levels were detected by staining the cells with CellROX Deep Red Reagent, ThermoFisher Scientific followed by flow cytometry.

## Histology

Lung tissues were fixed in formalin solution and coated with wax for sectioning. Sections were stained with Hematoxylin and Eosin (H and E) dyes and slides were scored for granulomas by analyz-ing under a light microscope.

## Antibodies

We used the following antibodies: CD4-PerCP (100538), CD8-APC/Cy7 (100713), CD69-FITC (104506), CD25-APC (101910), IFNγ-APC (505810), IL17-PE/Cy7 (506922), CD11b-APC/Cy7 (101226), CD11c-APC (117310), CD80-FITC (104706), MHCII-PE (107607), CD3-PacificBlue (100214), CD4-PE (100512), CD4-APC (100516) and CD4-FITC (100510) from Biolegend, USA.

SIRT2 (ab211033), β-actin (ab8227), acH3K18 (ab40888), α-tubulin (ab184613), acetyl-α-tubulin (ab179484), acetyl-NFκB p65 (ab19870), Alexa Fluor 647 (ab150075) and IgG-APC isotype control (ab232814) from Abcam.

ERK1/2 (9102), Phospho-ERK1/2 (9101), p-38 (9212) and phospho-p38 (4511) from Cell Signaling Technology.

## Statistical analysis

All mice experiments were done once with five mice per group per time point (n = 5). RNAseq was performed with three biological replicates from each condition. Significant differences were deter-mined by t-test or one-way ANOVA. $p < 0.05$ indicated statistically significant results.

## Acknowledgements

We acknowledge the support of the DBT-supported Tuberculosis Aerosol Challenge Facility at the International Centre for Genetic Engineering and Biotechnology (ICGEB), New Delhi, India, and their staff in accomplishing this work. AB and VPD are the recipients of DST-INSPIRE Faculty Fellowship, Department of Science and Technology, Government of India. We acknowledge the funding support from Department of Science and Technology, Government of India (NII/F-56/827/IFAD).

## Additional information

### Funding

| Funder | Grant reference number | Author |
| --- | --- | --- |
| Department of Science and Technology, Ministry of Science and Technology | DST/INSPIRE/04/2014/ 002012 | Ved Prakash Dwivedi |
| Department of Science and Technology, Ministry of Science and Technology | DST/INSPIRE/04/2014/ 002069 | Ashima Bhaskar |
| Department of Biotechnology , Ministry of Science and Technology | BT/PR13522/COE/34/27/ 2015 | Vinay Kumar Nandicoori |
| JC Bose Fellowship | JCB/2019/000015 | Vinay Kumar Nandicoori |

The funders had no role in study design, data collection and interpretation, or the decision to submit the work for publication.

### Author contributions

Ashima Bhaskar, Conceptualization, Resources, Data curation, Formal analysis, Supervision, Funding acquisition, Validation, Investigation, Visualization, Methodology, Writing - original draft, Project administration, Writing - review and editing; Santosh Kumar, Validation, Investigation; Mehak Zahoor Khan, Investigation, Writing - review and editing; Amit Singh, Conceptualization, Writing - review and editing; Ved Prakash Dwivedi, Resources, Investigation, Methodology, Writing - review and editing; Vinay Kumar Nandicoori, Resources, Writing - review and editing

### Author ORCIDs

Ashima Bhaskar https://orcid.org/0000-0002-7367-874X
Ved Prakash Dwivedi http://orcid.org/0000-0003-4321-2567
Vinay Kumar Nandicoori http://orcid.org/0000-0002-5682-4178

### Ethics

Animal experimentation: Animal experiments were carried out in accordance with the guidelines approved by the Animal Ethics Committee of National Institute of Immunology (NII, Approval ID: IAEC#409/16 & IAEC#462/18), New Delhi, India, International Centre for Genetic Engineering and Biotechnology (ICGEB, Approval ID: ICGEB/AH/2015/01/IMM-45), New Delhi, India and the Department of Biotechnology (DBT) Government of India. Mice were ethically sacrificed according to institutional and DBT regulations.

### Decision letter and Author response

Decision letter https://doi.org/10.7554/eLife.55415.sa1
Author response https://doi.org/10.7554/eLife.55415.sa2

## Additional files

### Supplementary files

- Transparent reporting form

## Data availability

All data generated or analysed during this study are included in the manuscript. Source data files have been provided for all the figures.

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
