## [Decision Letter]

**Acceptance summary:**

This work addresses the role of SIRT2 during *Mycobacterium tuberculosis* (*Mtb*) infection and how this can be exploited in new therapeutic approaches. These studies are important for our general understanding of *Mtb* pathogenesis as well as the development of new host-directed therapies.

**Decision letter after peer review:**

Thank you for submitting your article "Host Sirtuin 2 as an Immunotherapeutic Target against Tuberculosis" for consideration by *eLife*. Your article has been reviewed by three peer reviewers, and the evaluation has been overseen by a Reviewing Editor and Dominique Soldati-Favre as the Senior Editor.

The reviewers have discussed the reviews with one another and the Reviewing Editor has drafted this decision to help you prepare a revised submission.

Summary:

In this manuscript, the authors show that *Mtb* modulates NAD^+^ dependent histone deacetylase Sirtuin 2 (SIRT2) resulting in changes in histone acetylation, cellular signaling and transcriptional profile of the infected host macrophages and *Mtb* stimulated T-cells. Pharmacological inhibition of SIRT2 improved infection outcomes in *Mtb*-infected mice. Mice had reduced CFU, improved pathology and exhibited enhanced Th1/Th17 response. This is a nice study, but several key experiments are missing weakening the overall conclusions that inhibition of the SIRT2 pathway in T cells provides a novel therapeutic host target against *Mtb*. A compiled list of reviewer comments are included below, but as a summary, the major concerns that all reviewers agreed on were the manuscript's disjointed organization and focus, experiments needed to be repeated for rigor and reproducibility standards, missing key data (i.e. need to show CFU in Figure 10), missing key information on how experiments were performed, and the absence of an experiment demonstrating the role of T cell-specific-SIRT2 inhibition during *Mtb* infection.

Essential revisions:

1) Figure 10: A key measurement missing in this experiment is CFU. The authors show that the adoptively transferred T cells from SIRT2-deficient mice were polyfunctional. However, whether the adoptive transfer of these T cells affected the bacterial burden in the lungs was not determined. This is important since it is abundantly clear that polyfunctional T cells do not reflect the ability of the T cells to restrict intracellular *Mtb* growth.

2) Better quality and higher magnification H&E images should be included. Also, it would be helpful in evaluating pathological changes if the low magnification images of the entire lung lobe for all the mice are provided, maybe as supplementary data. Also, histopathological changes should be presented as quantitation of lung inflammation rather than presenting it as the number of granulomas.

3) Concerns with the flow of the paper, which was slightly disjointed – it starts with in vitro work showing macrophages are important for the SIRT2 effect, then onto consistent results in a mouse model treated with the same inhibitor then back to showing T cells matter when it turns out that the mouse result is not consistent with previous work. It might be smoother if the authors showed upfront that macrophages and T-cells were involved in altered SIRT2 and then go on to the in vivo results to dissect which aspect has the largest effect in vivo. Further, the authors explain the discrepancy between Cardoso et al., 2015 results (higher CFU early and then no effect by 60 days) being due to the myeloid-specific KO used in the previous study. The drug/inhibitor being used in the current study may indeed show the importance of cells other than myeloid as having a role, however, considering in the current study the authors spent the first few figures arguing for the SIRT2 effect as being specific to macrophages, I find it hard to reconcile the sudden shift in argument that it is independent of macrophages.

4) To demonstrate a role for T cells in vivo during Sirt2 inhibition, the authors could deplete T-cells in the mouse after infection (+/- inhibitor) to see if the protective effect (of lower CFU burden) is lost without T-cells.

5) The other hypothesis is that the inhibitor is not just against SIRT2 and may have off-target effects explaining the reduced bacterial growth in mice. This could be tested by having genetic (SIRT2 T cell specific KO mouse) targeting of SIRT2. Without genetic validation to show that the drug is working through Sirt2 specifically and Sirt2's important role is in T cells (although it may still be a combination of macs + T cells), the authors must be careful to not overstate their findings without this genetic data.

6) Figure 1A – Unclear why only SIRT1 and 2 data was shown, when the claim is that they are significantly upregulated/downregulated from a total of 18 HDACs. Why not show all or at least comment on the others not being altered? Also unclear why there are no error bars to show the variability of induction after infection. Depending on how the data was normalized, it may be hard to show standard deviation, but at least put confidence intervals so the reader can interpret variability.

7) Figure 1E – The timepoint for upregulation of SIRT2 expression from 1A was 24 hrs – in 1B the reduced bacterial growth was at 48 hours. Did the reduced bacterial growth also correlate with a reduction in SIRT2 after drug treatment at 48hours? Could this effect be seen at 24 hours?

8) Figure 1F – this is CFU data presented as % survival – it would be easier to readers to interpret if the average CFU for each replicate was shown instead. Also with Figure 1I.

9) The RNAseq data in Figure 3 was nice to show the transcriptional landscape of SIRT2 inhibition in cells but I’m not really sure what the difference in 3D and 3E is, except for different ways of showing that immune genes are altered after SIRT2 inhibition. Also unclear why only 3 upregulated and 2 downregulated genes were cherry-picked to validate. Were more checked but didn't validate? It might have been nice to validate some of genes of the pathways that were highlighted in D and E.

10) Considering all of the in vitro and ex vivo work in Figures 1-4 were done in C57BL/6 mice, it is unclear why the study then switched to BALB/c mice for the in vivo experiments (i.e. BALB/c are still relatively resistant to infection but have different H2 haplotypes and immune profiles) and then back to C57BL/6 mice for some of the T-cell coculture experiments. Could the discrepancy with in vivo results and past work simply be due to different host backgrounds being examined in vivo? Can the authors comments on that in the Discussion?

11) In Figure 8 when infected mouse spleens are immunoprofiled, it is unclear why the authors chose the spleen and not the lung. It has been shown that there are organ-specific differences in protective immunity (Sakai, 2016). Was this experiment back to BALB/c mice again? This should be made clear in the figure legend.

12) There is no mention in the Introduction that genetic Sirt2 deletion in myeloid cells was shown to increase *Mtb* load in the lungs of mice early but not at later timepoints (Cardoso et al., 2015) or that SIRT2 deficiency enhances bacterial phagocytosis as a mechanism explaining increased survival in chronic staph infection (Ciarlo, 2017). These references are mentioned later in the Results and Discussion when the results are discrepant from previous studies, but it might be nice to mention these on the front to so the reader is aware there has been past work in this area (at the moment, the Introduction seems to be pitched that no one has looked at sirtuins in *Mtb* before).

13) Some hyperbole throughout manuscript – for e.g. "to our astonishment"; "brilliant strategies" could be toned down.

14) What was the sex of the mice used in vitro and in vivo? Note sex differences have been observed in both human and mouse studies. For the field standard and reproducibility, it would be helpful to include the mouse sex in the Materials and methods.

15) Several of the in vivo experiments (Figure 8, 9 and 10), the adoptive transfer experiment and splenocyte flow cytometry experiment were only performed once with 3 mice – for rigor and reproducibility, the authors should repeat these experiments for independent biological replicates.

[Editors' note: further revisions were suggested prior to acceptance, as described below.]

Thank you for resubmitting your work entitled "Host Sirtuin 2 as an Immunotherapeutic Target against Tuberculosis" for further consideration by *eLife*. Your revised article has been evaluated by Dominique Soldati-Favre (Senior Editor) and a Reviewing Editor.

The manuscript has been improved but there are some remaining issues that need to be addressed before acceptance, as outlined below:

The reviewers all agree that the revised manuscript is greatly improved, and additional data have been provided to mostly address the original concerns. There were just a few remaining points to be addressed and included:

1) The authors should describe what the control cells are (Figure 10 G and H) in the adoptive transfer experiment. Are they cells from infected but not AGK2 infected mice?

2) Please add a discussion of the increase in acetylated-tubulin in infected (Cnt) cells compared to uninfected (Cnt) cells and the similar increase in infected (AGK2) cells compared to uninfected (AGK2) cells in Figure 2E.

3) For Figure 4, Figure 6E-J, please include data from both experiments.

4) Subsection “SIRT2 deacetylates NFκB p65 in *Mtb* specific T cells”, fourth paragraph should be read as – SIRT2 "inhibition by AGK2" modulated the acetylation.

5) Figure 7 – There are issues with the gating strategy of the lung APCs. The first gate is labeled as lymphocytes, but that is not an accurate label for that population. In general, how the authors decided on their gates is not clear and does not appear to match the pattern of cells on the plots. The boxes do not contain distinct populations and often cut across populations. Were fluorescence minus one (FMO) controls used to determine where to make the cutoff for each marker? Can authors provide details regarding what they think each population is, in addition to defining the populations based on markers?

6) Please confirm if the mouse experiments (Figure 7 – 10) were done more than once, and if they were only done once, please add a replicate experiment.

---

## [Author Response]

Essential revisions:1) Figure 10: A key measurement missing in this experiment is CFU. The authors show that the adoptively transferred T cells from SIRT2-deficient mice were polyfunctional. However, whether the adoptive transfer of these T cells affected the bacterial burden in the lungs was not determined. This is important since it is abundantly clear that polyfunctional T cells do not reflect the ability of the T cells to restrict intracellular Mtb growth.

We thank the editor and the reviewers for the insightful comments that have improved both the content and the presentation of our work. We agree with the reviewers. To investigate whether the T cells isolated from AGK2 treated infected animals transfer protective immunity against *Mtb*, CD4^+^ and CD8^+^ T cells from control and AGK2 treated infected mice were transferred into naive immuno-deficient *Rag1^-/-^* mice followed by low dose infection (~110 CFU/lung) with aerosolized *Mtb* and CFU enumeration after 21 days post infection. T cells from AGK2 treated animals were more efficient in imparting protection to the *Rag1^-/-^* mice. The data is provided in Figure 10G and H.

2) Better quality and higher magnification H&E images should be included. Also, it would be helpful in evaluating pathological changes if the low magnification images of the entire lung lobe for all the mice are provided, maybe as supplementary data. Also, histopathological changes should be presented as quantitation of lung inflammation rather than presenting it as the number of granulomas.

We thank the reviewers for the comments. In the revised manuscript, we have included the representative 10X, 20X and 40X magnification images of the lungs (Figure 6B). The quantification of the lung inflammation is done in accordance with the previous literature (Soni et al., 2015). The table describing the number and nature of granulomas has now been included in the manuscript (Figure 6C).

3) Concerns with the flow of the paper, which was slightly disjointed – it starts with in vitro work showing macrophages are important for the SIRT2 effect, then onto consistent results in a mouse model treated with the same inhibitor then back to showing T cells matter when it turns out that the mouse result is not consistent with previous work. It might be smoother if the authors showed upfront that macrophages and T-cells were involved in altered SIRT2 and then go on to the in vivo results to dissect which aspect has the largest effect in vivo. Further, the authors explain the discrepancy between Cardoso et al., 2015 results (higher CFU early and then no effect by 60 days) being due to the myeloid-specific KO used in the previous study. The drug/inhibitor being used in the current study may indeed show the importance of cells other than myeloid as having a role, however, considering in the current study the authors spent the first few figures arguing for the SIRT2 effect as being specific to macrophages, I find it hard to reconcile the sudden shift in argument that it is independent of macrophages.

We appreciate the comments concerning the flow of the manuscript. In the revised manuscript we have reorganized the figures in order to segregate the ex vivo and in vivo work more effectively.

We agree with the reviewers that the first two figures of the manuscript describe the importance of SIRT2 inhibition in macrophages in restricting intracellular bacterial growth. Our rationale behind starting the manuscript with the macrophage infection experiments was based on the fact that the macrophages are one of the very first cells to encounter *Mtb* inside the host. The transcriptome analysis of macrophages following *Mtb* infection and SIRT2 inhibition revealed important insights into a plausible role of SIRT2 in modulating T cell responses during *Mtb* infection and hence we went further to investigate the T cell mediated responses after SIRT2 inhibition.

We do concur that our ex vivo and in vivo results do not follow the same trend as described by Cardoso et al. in 2015 where myeloid-specific SIRT2 KO mice displayed higher bacterial burden at early time points with no effect at 60 days post infection. To address this disparity, we hypothesized that it is the inhibition of SIRT2 in cells other than macrophages is critical for containing the *Mtb* infection. To validate the hypothesis, we performed experiments in *Rag1^-/-^* mice that lack mature B and T cells. AGK2 treatment in these immuno-deficient mice did not alter the bacterial survival as compared to the control group (Figure 6H-J) indicating that SIRT2 inhibition in T cells is important to restrict *Mtb* growth inside the host. However, since macrophages are essential for activating T cell mediated immunity, we do not believe that the phenotype is independent of macrophages.

4) To demonstrate a role for T cells in vivo during Sirt2 inhibition, the authors could deplete T-cells in the mouse after infection (+/- inhibitor) to see if the protective effect (of lower CFU burden) is lost without T-cells.

We would like to thank the reviewers for their suggestion. To demonstrate the importance of SIRT2 inhibition in T cells, we used *Rag1^-/-^* mice which lack mature T cells. As discussed in the previous response, the effect of AGK2 treatment in terms of reduced bacterial survival is lost in these immuno-deficient mice (Figure 6H-J) indicating that inhibition of SIRT2 in T cells is important to restrict *Mtb* growth inside the host.

5) The other hypothesis is that the inhibitor is not just against SIRT2 and may have off-target effects explaining the reduced bacterial growth in mice. This could be tested by having genetic (SIRT2 T cell specific KO mouse) targeting of SIRT2. Without genetic validation to show that the drug is working through Sirt2 specifically and Sirt2's important role is in T cells (although it may still be a combination of macs + T cells), the authors must be careful to not overstate their findings without this genetic data.

We completely agree with the reviewers on the possible unknown off-target effects of AGK2. The experiments in *Rag1^-/-^* mice (Figure 6H-J) to an extent suggest that the observed phenotype is likely not due to do the off-target effects of AGK2. Nonetheless, we agree that in the absence of genetic validation, we should be careful not to overstate the findings. Thus, we have avoided making any overstatements in the revised manuscript.

6) Figure 1A – Unclear why only SIRT1 and 2 data was shown, when the claim is that they are significantly upregulated/downregulated from a total of 18 HDACs. Why not show all or at least comment on the others not being altered? Also unclear why there are no error bars to show the variability of induction after infection. Depending on how the data was normalized, it may be hard to show standard deviation, but at least put confidence intervals so the reader can interpret variability.

We would like to apologize for not commenting on all the HDACs. In the revised manuscript, bar graph in Figure 1A is replaced by a table detailing the expression of all the HDACs which are significantly altered in THP1 cells at 24h post infection (SIRT1, SIRT2, HDAC6 and HDAC9). The data was analysed with the help of GEO2R (Gene Expression Omnibus series accession number GSE65714).

7) Figure 1E – The timepoint for upregulation of SIRT2 expression from 1A was 24 hrs – in 1B the reduced bacterial growth was at 48 hours. Did the reduced bacterial growth also correlate with a reduction in SIRT2 after drug treatment at 48hours? Could this effect be seen at 24 hours?

AGK2 is a specific and a reversible inhibitor of SIRT2 activity. It is known that AGK2 targets the nicotinamide binding site of SIRT2 (Outeiro et al., 2007). However, our data suggests that AGK2 treatment during *Mtb* infection also leads to the reduction of SIRT2 mRNA levels. We believe that the reduced bacterial survival after 48h of AGK2 treatment is because of the inhibition of SIRT2 activity as well as a reduction in SIRT2 levels. Similar mycobacterial growth restriction was observed as early as 24h post treatment with AGK2. The data is provided in Figure 1E.

8) Figure 1F – this is CFU data presented as % survival – it would be easier to readers to interpret if the average CFU for each replicate was shown instead. Also with Figure 1I.

The CFU data was presented as % survival wrt the 0h time point to negate the effect of any variation in the uptake of the bacilli due to the pre-treatment with AGK2. In the revised manuscript, the data in Figure 1F and 1I has been replaced by the average CFU for each replicate at 0h and 48h post infection.

9) The RNAseq data in Figure 3 was nice to show the transcriptional landscape of SIRT2 inhibition in cells but I’m not really sure what the difference in 3D and 3E is, except for different ways of showing that immune genes are altered after SIRT2 inhibition. Also unclear why only 3 upregulated and 2 downregulated genes were cherry-picked to validate. Were more checked but didn't validate? It might have been nice to validate some of genes of the pathways that were highlighted in D and E.

We have rearranged the figure. Earlier Figure 3D depicted the KEGG pathway analysis of all the SIRT2 dependent genes whereas Figure 3E represented the gene ontology immune system analysis on SIRT2 dependent upregulated genes. However, to avoid confusion we have removed Figure 3E in the revised version. We have also included RT-PCR on the candidate genes which belong to key pathways such as antigen presentation and processing, Th17 differentiation, T cell activation, NFκB signalling and PI3K-Akt signalling (Figure 3D and 3E).

10) Considering all of the in vitro and ex vivo work in Figures 1-4 were done in C57BL/6 mice, it is unclear why the study then switched to BALB/c mice for the in vivo experiments (i.e. BALB/c are still relatively resistant to infection but have different H2 haplotypes and immune profiles) and then back to C57BL/6 mice for some of the T-cell coculture experiments. Could the discrepancy with in vivo results and past work simply be due to different host backgrounds being examined in vivo? Can the authors comments on that in the Discussion?

We thank the reviewers for the comments. In order to maintain the consistency in the ex vivo and in vivo work, we have now repeated the in vivo experiments in C57BL/6 mice. The *Rag1^-/-^* mice used in the study are also in C57BL/6 background.

11) In Figure 8 when infected mouse spleens are immunoprofiled, it is unclear why the authors chose the spleen and not the lung. It has been shown that there are organ-specific differences in protective immunity (Sakai, 2016). Was this experiment back to BALB/c mice again? This should be made clear in the figure legend.

To address the concern, we have included the immune profiling data of the lungs of infected C57BL/6 mice (Figure 7A-E, Figure 8A-G and Figure 9A-G) in the revised manuscript.

12) There is no mention in the Introduction that genetic Sirt2 deletion in myeloid cells was shown to increase Mtb load in the lungs of mice early but not at later timepoints (Cardoso et al., 2015) or that SIRT2 deficiency enhances bacterial phagocytosis as a mechanism explaining increased survival in chronic staph infection (Ciarlo, 2017). These references are mentioned later in the Results and Discussion when the results are discrepant from previous studies, but it might be nice to mention these on the front to so the reader is aware there has been past work in this area (at the moment, the Introduction seems to be pitched that no one has looked at sirtuins in Mtb before).

We thank the reviewers for the insight. In the revised manuscript, we have modified the Introduction by adding these references and a few more which talk about sirtuins and HDACs during bacterial infections.

13) Some hyperbole throughout manuscript – for e.g. "to our astonishment"; "brilliant strategies" could be toned down.

We appreciate the reviewer’s concerns. The point is well noted, and necessary changes have been made.

14) What was the sex of the mice used in vitro and in vivo? Note sex differences have been observed in both human and mouse studies. For the field standard and reproducibility, it would be helpful to include the mouse sex in the Materials and methods.

In the revised manuscript, we have included the details about the sex of mice used in the ex vivo and in vivo studies.

15) Several of the in vivo experiments (Figure 8, 9 and 10), the adoptive transfer experiment and splenocyte flow cytometry experiment were only performed once with 3 mice – for rigor and reproducibility, the authors should repeat these experiments for independent biological replicates.

We have repeated these experiments in C57BL/6 mice (5 mice per group).

[Editors' note: further revisions were suggested prior to acceptance, as described below.]

[…]1) The authors should describe what the control cells are (Figure 10 G and H) in the adoptive transfer experiment. Are they cells from infected but not AGK2 infected mice?

The control cells (Cnt) in Figure 10G and 10H represent the T cells isolated from *Mtb* infected animals which were not treated with AGK2 while AGK2 represents the cells isolated from *Mtb* infected and AGK2 treated mice. We have added this information in the Results section of the revised manuscript.

2) Please add a discussion of the increase in acetylated-tubulin in infected (Cnt) cells compared to uninfected (Cnt) cells and the similar increase in infected (AGK2) cells compared to uninfected (AGK2) cells in Figure 2 E.

There is enhanced nuclear localization of SIRT2 during *Mtb* infection which leaves very little protein in the cytosol and as a result there is an overall increase in the acetylated-tubulin levels in *Mtb* infected cells as compared to the uninfected macrophages. We have added this discussion in the Results section of the revised manuscript.

3) For Figure 4, Figure 6E-J, please include data from both experiments.

The figures in the revised version include the data from both the experiments.

4) Subsection “SIRT2 deacetylates NFκB p65 in Mtb specific T cells”, fourth paragraph should be read as – SIRT2 "inhibition by AGK2" modulated the acetylation.

We have made the asked changes in the revised manuscript.

5) Figure 7 – There are issues with the gating strategy of the lung APCs. The first gate is labeled as lymphocytes, but that is not an accurate label for that population. In general, how the authors decided on their gates is not clear and does not appear to match the pattern of cells on the plots. The boxes do not contain distinct populations and often cut across populations. Were fluorescence minus one (FMO) controls used to determine where to make the cutoff for each marker? Can authors provide details regarding what they think each population is, in addition to defining the populations based on markers?

We would like to thank the reviewers for their insights. We would like to apologize for the mislabelling of the population in dot plot. By default, FlowJo software names the first population to be gated as lymphocytes. In the revised version, we have changed the name of the first gate in Figure 7-9. Since in Figure 7-9, we are analysing the total lung and spleen cells, the first gate was decided based on the distinct population by adjusting FSC and SSC voltages followed by doublet exclusion gates. Yes, we use FMO controls for the gating. We also use unstained and single colour controls to apply the gates over the population wherever it is necessary. Further, we make use of contour plots to mark the populations whenever they are not clearly distinct in the dot plots. Finally, we apply the same gates across the samples to be analysed.

6) Please confirm if the mouse experiments (Figure 7 – 10) were done more than once, and if they were only done once, please add a replicate experiment.

Yes, the mice experiments in Figure 6-10 were done more than once. The CFU data from two biological replicates have been added in Figure 6E-J and Figure 10H. We apologize for not including the biological replicate in the earlier version.

References

Soni V, Upadhayay S, Suryadevara P, Samla G, Singh A, Yogeeswari P, Sriram D, Nandicoori VK (2015) Depletion of *M. tuberculosis* GlmU from Infected Murine Lungs Effects the Clearance of the Pathogen. PLoS Pathog. 11(10):e1005235.